# A Theory of Unsupervised Translation Motivated by Understanding Animal Communication

**Shafi Goldwasser**[*]
UC Berkeley & Project CETI
shafi.goldwasser@berkeley.edu

**David F. Gruber**[*]
Project CETI
david@projectceti.org

**Adam Tauman Kalai**[*]
Microsoft Research & Project CETI
adam@kal.ai

**Orr Paradise**[*]
UC Berkeley & Project CETI
orrp@eecs.berkeley.edu

## Abstract

Neural networks are capable of translating between languages—in some cases even between two languages where there is little or no access to parallel translations, in what is known as Unsupervised Machine Translation (UMT). Given this progress, it is intriguing to ask whether machine learning tools can ultimately enable understanding animal communication, particularly that of highly intelligent animals. We propose a theoretical framework for analyzing UMT when no parallel translations are available and when it cannot be assumed that the source and target corpora address related subject domains or posses similar linguistic structure. We exemplify this theory with two stylized models of language, for which our framework provides bounds on necessary sample complexity; the bounds are formally proven and experimentally verified on synthetic data. These bounds show that the error rates are inversely related to the language complexity and amount of common ground. This suggests that unsupervised translation of animal communication may be feasible if the communication system is sufficiently complex.

## 1 Introduction

Recent interest in translating animal communication [2, 3, 9] has been motivated by breakthrough performance of Language Models (LMs). Empirical work has succeeded in unsupervised translation between human-language pairs such as English–French [23, 5] and programming languages such as Python–Java [33]. Key to this feasibility seems to be the fact that language statistics, captured by a LM (a probability distribution over text), encapsulate more than just grammar. For example, even though both are grammatically correct, *The calf nursed from its mother* is more than 1,000 times more likely than *The calf nursed from its father*.[2]

Given this remarkable progress, it is natural to ask whether it is possible to collect and analyze animal communication data, aiming towards translating animal communication to a human language description. This is particularly interesting when the source language may be of highly social and intelligent animals, such as whales, and the target language is a human language, such as English.

**Challenges.** The first and most basic challenge is *understanding the goal*, a question with a rich history of philosophical debate [38]. To define the goal, we consider a hypothetical ground-truth translator. As a thought experiment, consider a "mermaid" fluent in English and the source language

---

[*]Authors listed alphabetically.
[2]Probabilities computed using the GPT-3 API https://openai.com/api/ text-davinci-02 model.

37th Conference on Neural Information Processing Systems (NeurIPS 2023).

| A | B | C |
|---|---|---|
| *Have you seen any orcas today? I just got back from the reef.* $p(A_1) \approx 10^{-18}$ | *Have you seen mom? I just returned from the ocean basin.* $p(B_1) \approx 10^{-18}$ | *Hat out hat dsjgh!!!* $p(C_1) \approx 10^{-48}$ |
| *At the reef, there were a lot of sea turtles.* $p(A) \approx 10^{-22}$ | *At the reef, there were a lot of sea turtles.* $p(B) \approx 10^{-26}$ | *bicycle OMG and.* $p(C) \approx 10^{-72}$ |

Figure 1: LMs identify incoherent text. The probabilities of three two-paragraph texts computed using the GPT-3 API. The probabilities of just the first paragraphs $A_1, B_1, C_1$ are also shown. Although $p(A_1) \approx p(B_1)$ and the second paragraphs of $A$ and $B$ are identical, overall $p(A) \gg p(B)$ due to coherence between the paragraphs. $C$ is gibberish.

(e.g. sperm whale communication). Such a mermaid could translate whale vocalizations that English naturally expresses. An immediate worry arises: what about communications that the source language may have about topics for which English has no specific words? For example, sperm whales have a sonar sense which they use to perform echolocation. In that case, lacking a better alternative, the mermaid may translate such a conversation as *(something about echolocation)*.[3]

Thus, we formally define the goal to be to achieve translations similar to those that would be output by a hypothetical ground-truth translator. While this does not guarantee functional utility, it brings the general task of unsupervised translation and the specific task of understanding animal communication into the familiar territory of supervised translation, where one can use existing error metrics to define (hypothetical) error rates.

The second challenge is that animal communication is unlikely to share much, if any, linguistic structure with human languages. Indeed, our theory will make no assumption on the source language other than that it is presented in a textual format. That said, one of our instantiations of the general theory (the knowledge graph) shows that translation is easier between compositional languages.

The third challenge is *domain gap*, i.e., that ideal translations of animal communications into English would be semantically different from existing English text, and we have no precise prior model of this semantic content. (In contrast, the distribution of English translations of French text would resemble the distribution of English text.) Simply put: whales do not "talk" about smartphones. Instead, we assume the existence of a *broad prior* that models plausible English translations of animal communication. LMs assign likelihood to an input text based not only on grammatical correctness, but also on agreement with the training data. In particular, LMs trained on massive and diverse data, including some capturing facts about animals, may be able to reason about the plausibility of a candidate translation. See Figure 1 and the discussion in Appendix H.

## 1.1 Framework and results

A translator[4] is a function $f \colon \mathcal{X} \to \mathcal{Y}$ that translates source text $x \in \mathcal{X}$ into the target language $f(x) \in \mathcal{Y}$. We focus on the easier-to-analyze case of *lossless* translation, where $f$ is invertible (one-to-one) denoted by $f_\theta \colon \mathcal{X} \hookrightarrow \mathcal{Y}$. See Appendix I.1 for an extension to lossy translation.

We will consider a parameterized family of translators $\{f_\theta \colon \mathcal{X} \to \mathcal{Y}\}_{\theta \in \Theta}$, with the goal being to learn the parameters $\theta \in \Theta$ of the most accurate translator. Accuracy (defined shortly) is measured with respect to a hypothetical *ground-truth* translator denoted by $f_\star \colon \mathcal{X} \to \mathcal{Y}$. We make a *realizability* assumption that the ground-truth translator can be represented in our family, i.e., $\star \in \Theta$.

The source language is defined as a distribution $\mu$ over $x \in \mathcal{X}$, where $\mu(x)$ is the likelihood that text $x$ occurs in the source language. The error of a model $\theta \in \Theta$ will be measured in terms of $\mathrm{err}(\theta) := \Pr_{x \sim \mu}[f_\theta(x) \neq f_\star(x)]$, or at times a general bounded loss function $\mathcal{L}(\theta)$. Given $\mu$ and $f_\star$, it will be useful to consider the *translated language distribution* $\tau$ over $\mathcal{Y}$ by taking $f_\star(x)$ for $x \sim \mu$.

In the case of similar source and target domains, one may assume that the target language distribution $\nu$ over $\mathcal{Y}$ is close to $\tau$. This is a common intuition given for the "magic" behind why UMT sometimes

---

[3] A better description might be possible. Consider the fact that some people who are congenitally blind comprehend vision-related verbs as accurately as sighted people [8].

[4] In this work, a *translator* refers to the function $f \colon \mathcal{X} \to \mathcal{Y}$ while *translation* refers to an output $y = f(x)$.

works: for complex asymmetric distributions, there may a nearly unique transformation in $\{f_\theta\}_{\theta\in\Theta}$ that maps $\mu$ to something close $\nu$ (namely $f_\star$ which maps $\mu$ to $\tau$). A common approach in UMT is to embed source and target text as high-dimensional vectors and learn a *low-complexity* transformation, e.g., a rotation between these Euclidean spaces. Similarly, translator complexity will also play an important role in our analysis.

**Priors.** Rather than assuming that the target distribution $\nu$ is similar to the translated distribution $\tau$, we will instead assume access to a broad prior $\rho$ over $\mathcal{Y}$ meant to capture how plausible a translation $y$ is, with larger $\rho(y)$ indicating more natural and plausible translation. Appendix H discusses one way a prior oracle can be created, starting with an LM $\approx \nu$ learned from many examples in the target domain, and combined with a prompt, in the target language, describing the source domain.

We define the problem of *unsupervised machine translation (with a prior)* to be finding an accurate $\theta \in \Theta$ given $m$ iid unlabeled source texts $x_1, \ldots, x_m \sim \mu$ and oracle access to prior $\rho$.

**MLE.** Our focus is on the Maximum-Likelihood Estimator (MLE), which selects model parameters $\theta \in \Theta$ that (approximately) maximize the likelihood of translations $\prod_i \rho\big(f_\theta(x_i)\big)$.

**Definition 1.1** (MLE)**.** *Given input a translator family* $\{f_\theta \colon \mathcal{X} \to \mathcal{Y}\}_{\theta\in\Theta}$*, samples* $x_1, \ldots, x_m \in \mathcal{X}$ *and a distribution* $\rho$ *over* $\mathcal{Y}$*, the* MLE *outputs*

$$\mathrm{MLE}^\rho(x_1, x_2, \ldots, x_m) := \operatorname*{argmin}_{\theta\in\Theta} \sum_{i=1}^{m} \log \frac{1}{\rho(f_\theta(x_i))}$$

*If multiple* $\theta$ *have equal empirical loss, it breaks ties, say, lexicographically.*

We note that heuristics for MLE have proven extremely successful in training the breakthrough LMs, even though MLE optimization is intractable in the worst case.

Next, we analyze the efficacy of MLE in two complementary models of language: one that is highly structured (requiring compositional language) and one that is completely unstructured. These analyses both make strong assumptions on the target language, but make few assumptions about the source language itself. In both cases, the source distributions are uniform over subsets of $\mathcal{X}$, which (informally) is the "difficult" case for UMT as the learner cannot benefit from similarity in text frequency across languages. Both models are parameterized by the amount of "common ground" between the source language and the prior, and both are randomized. Note that these models are *not* intended to accurately capture natural language. Rather, they illustrate how our theory can be used to study the effect of language similarity and complexity on data requirements for UMT.

**Knowledge graph model.** Our first model consists of a pair of related *knowledge graphs*, in which edges encode knowledge of binary relations. Each edge yields a text that described the knowledge it encodes. For example, in Figure 2 edges encode which animal $A$ eats which other animal $B$, and text is derived as a simple description $A$ *eats* $B$.[5]

Formally, there are two Erdős–Rényi random digraphs. The target graph is assumed to contain $n$ nodes, while the source graph has $r \le n$ nodes corresponding to an (unknown) subset of the $n$ target nodes. The model has two parameters: the average degree $d$ in the (randomly-generated) target language graph, and the agreement $\alpha \in (0, 1]$ between the source and the target graphs. Here $\alpha = 1$ is complete agreement on edges in the subgraph, while $\alpha = 0$ is complete independence. We assume the languages use a *compositional* encoding for edges, meaning that they encode a given edge by encoding both nodes, so we may consider only $|\Theta| = n!/(n-r)!$ translators consistently mapping the $r$ source nodes to the $n$ target nodes, which is many fewer than the number of functions $f : \mathcal{X} \to \mathcal{Y}$ mapping the $|\mathcal{X}| = O(r^2)$ source edges into the $\mathcal{Y} = O(n^2)$ target edges. Human languages as well as the communication systems of several animals are known to be compositional [39].[6]

We analyze how the error of the learned translator depends on this "common knowledge" $\alpha$:

**Theorem 1.2** (Theorem 3.2, simplified)**.** *Consider a source language* $\mu$ *and a prior* $\rho$ *generated by the knowledge graph model over* $r$ *source nodes,* $n$ *target nodes, average degree* $d$ *and agreement*

---

[5]In a future work, it would be interesting to consider $k$-ary relations (hypergraphs) or multiple relations.

[6]A system is *compositional* if the meaning of an expression is determined by the meaning of its parts.

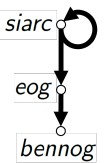

| Welsh sentences | English sentences |
| --- | --- |
| *bwytaodd siarc siarc.* | *sharks eat salmon.* |
| *bwytaodd eog bennog.* | *salmon eat herring.* |
| *bwytaodd siarc eog.* | *sharks eat sharks.* |
| | *herring eat mysidacea.* |

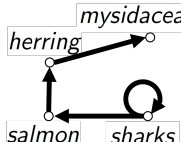

Figure 2: An illustration of the knowledge graph model. In this example, the Welsh graph is an exact subgraph of the English knowledge graph, but our model allows for differences.

*parameter $\alpha$. Then, with at least $99\%$ probability, when given $m$ source sentences $x_1, \ldots, x_m$ and access to a prior $\rho$,* MLE *outputs a translator $\hat{\theta}$ with error*

$$\mathrm{err}(\hat{\theta}) \leq O\left(\frac{\log n}{\alpha^2 d} + \frac{1}{\alpha}\sqrt{\frac{r \log n}{m}}\right).$$

The second term decreases to 0 at a $O(m^{-1/2})$ rate, similar to (noisy) generalization bounds [30]. Note that the first term does not decrease with the number of samples. The average degree $d$ is a rough model of language complexity capturing per-node knowledge, while the agreement parameter $\alpha$ captures the amount of common ground. Thus, more complex languages can be translated (within a given error rate) with less common ground. Even with $m = \infty$ source data, there could still be errors in the mapping. For instance, there could be multiple triangles in the source and target graphs that lead to ambiguities. However, for complex knowledge relations (degree $d \gg 1/\alpha^2$), there will be few such ambiguities.

Figure 2 illustrates an example of four English sentences and three sentences (corresponding to an unknown subset of three of the English sentences) in Welsh. For UMT, one might hypothesize that *bwytaodd* means *eat* because they both appear in every sentence. One might predict that *siarc* means *shark* because the word *siarc* appears twice in a single Welsh sentence and only the word shark appears twice in an English sentence. Next, note that *eog* may mean *salmon* because they are the only other words occurring with *siarc* and *shark*. Similar logic suggests that *bennog* means *herring*. Furthermore, the word order is consistently permuted, with subject-verb-object in English and verb-subject-object in Welsh. This translation is indeed roughly correct. This information is encoded in the directed graphs as shown, where each node corresponds to an animal species and an edge between two nodes is present if the one species eats the other.

**"Common nonsense" model.**   The second model, the *common nonsense* model, assumes no linguistic structure on the source language. Here, we set out to capture the fact that the translated language $\tau = \mu \circ f_\star$ and the prior $\rho$ share some common ground through the fact that the laws of nature may exclude common "nonsense" outside both distributions' support.

Earlier work has justified *alignment* for UMT under the intuition that the target language distribution $\nu$ is approximated by a nearly unique simple transformation, e.g., a rotation, of the source distribution $\mu$. However, for a prior $\rho$, our work suggests that UMT may also be possible if there is nearly a unique simple transformation that maps $\tau$ so that it is *contained* in $\rho$. Figure 3 illustrates such a nearly unique rotation—the UMT "puzzle" of finding a transformation $f_\theta$ of $\mu$ which is contained within $\rho$ is subtly different from finding an alignment.

In the common nonsense model, $\tau$ and $\rho$ are uniform over arbitrary sets $T \subseteq P \subseteq \mathcal{Y}$ from which a common $\alpha \in (0, 1/2]$ fraction of text is removed (hence the name "common nonsense"). Specifically, $\tau$ and $\rho$ are defined to be uniform over $\tilde{T} = T \setminus S$, $\tilde{P} = P \setminus S$, respectively, for a set $S$ sampled by including each $y \in \mathcal{Y}$ with probability $\alpha$.

We analyze the error of the learned translator as a function of the amount of common nonsense:

**Theorem 1.3** (Theorem 3.4, simplified).  *Consider source language $\mu$ and a prior $\rho$ generated by the common nonsense model over $|T|$ source texts and common-nonsense parameter $\alpha$, and a translator family parameterized by $|\Theta|$. Then, with at least $99\%$ probability, when given $m$ source sentences $x_1, \ldots, x_m$ and access to a prior $\rho$,* MLE *outputs a translator $\hat{\theta}$ with error*

$$\mathrm{err}(\hat{\theta}) := \Pr_{x \in X}[f_\theta(x) \neq f_\star(x)] = O\left(\frac{\ln|\Theta|}{\alpha \min(m, |T|)}\right).$$

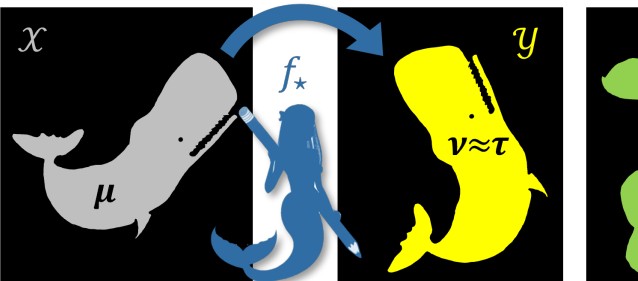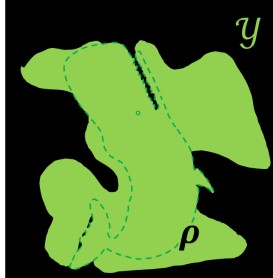

Figure 3: The previous intuition behind UMT has the distributions of target language $\nu$ (middle) close to ground-truth translations $\tau$, which is assumed to be a low-complexity transformation (in this example a rotation) of the source language $\mu$ (left). When source and target are not aligned, restricting to *prior* $\rho$ region (right) allows for translation, as long as there are enough "nonsense" texts (black regions) so that there is a nearly unique rotation of $\mu$ that is contained in $\rho$. For example, both distributions may assign negligible probability to nonsensical texts such as *I died 3 times tomorrow*. (In this toy example, $\mu$ is uniform over a two-dimensional shape that happens to look like a whale.)

Theorem 3.5 gives a nearly matching lower bound. Let us unpack the relevant quantities. First, we think of $\alpha$ as measuring the amount of agreement or common ground required, which might be a small constant. Second, note that $|T|$ is a coarse measure of the complexity of the source language, which requires a total of $\tilde{O}(|T|)$ bits to encode. Thus, the bound suggests that accurate UMT requires the translator to be simple, with a description length that is an $\alpha$-factor of the language description length, and again $\alpha$ captures the agreement between $\tau, \rho$. Thus, even with limited common ground, one may be able to translate from a source language that is sufficiently complex. Third, for simplicity, we require $\mathcal{X}, \mathcal{Y} \subset \{0, 1\}^*$ to be finite sets of binary strings, so WLOG $\Theta$ may be also assumed to be finite. Thus, $\log_2 |\Theta|$ is the *description length*, a coarse but useful complexity measure that equals the number of bits required to describe any model. (Neural network parameters can be encoded using a constant number of bits per parameter.) Appendix I.2 discusses how this can be generalized to continuous parameters.

Importantly, we note that (supervised) neural machine translators typically use far fewer parameters than LMs.[7] To see why, consider the example of the nursing calf (page 1) and the fact that a translator needs not know that calves nurse from mothers. On the other hand, such knowledge is essential to generate realistic text. Similarly, generating realistic text requires maintaining coherence between paragraphs, while translation can often be done at the paragraph level.

As a warm-up, we include a simplified version of the common nonsense model, called the *tree-based model* (Appendix B.1), in which texts are constructed word-by-word based on a tree structure.

**Comparison to supervised classification.** Consider the dependency on $m$, the number of training examples. Note that the classic Occam bound $O\left(\frac{1}{m} \log |\Theta|\right)$ is what one gets for noiseless supervised classification, that is, when one is also given labels $y_i = f_\star(x_i)$ at training time, which is similar to Theorem 1.3, and give $\tilde{O}(m^{-1/2})$ bounds for noisy classification as in Theorem 1.2. Furthermore, these bounds apply to translation, which can be viewed as a special case of classification with *many* classes $\mathcal{Y}$. Thus, in both cases, the data dependency on $m$ is quite similar to that of classification.

**Experiments.** We validate our theorems generating synthetic data from randomly-generated languages according to each model, and evaluating translator error as a function of the number of samples and amount of common ground. The knowledge graph model (Figure 4, left) is used to generate a source graph (language) on $r = 9$ nodes to a target graph (language) on $n = 10$ nodes and average degree $d \approx 5$, while varying the agreement parameter $\alpha$. We also vary $r$ (Figure 4, right) supporting our main message: more complex languages can be translated more accurately. For the common nonsense model (Figure 5) we simulate translation of a source language of size $|T| = 10^5$ while varying the fraction of common nonsense $\alpha$. Appendix E contains details and code.

---

[7]For example, a multilingual model achieves state-of-the-art performance using only 5 billion parameters [37], compared to 175 billion for GPT-3 [11].

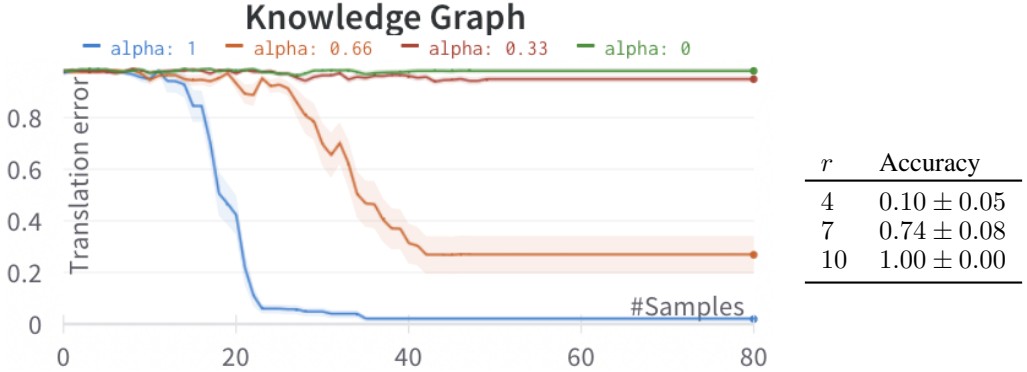

| $r$ | Accuracy |
|-----|-----------------|
| 4 | $0.10 \pm 0.05$ |
| 7 | $0.74 \pm 0.08$ |
| 10 | $1.00 \pm 0.00$ |

Figure 4: Knowledge Graph model experiments, each run on twenty seeds with standard errors shown. Left: error of the top-scoring translator vs. number of source samples $m$. Right: effect of source language complexity (number of source nodes $r$) on translator accuracy in the knowledge graph model. We report the accuracy of the top-scoring translator after all source edges were input to the learning algorithm, i.e., as the number of samples $m \to \infty$.

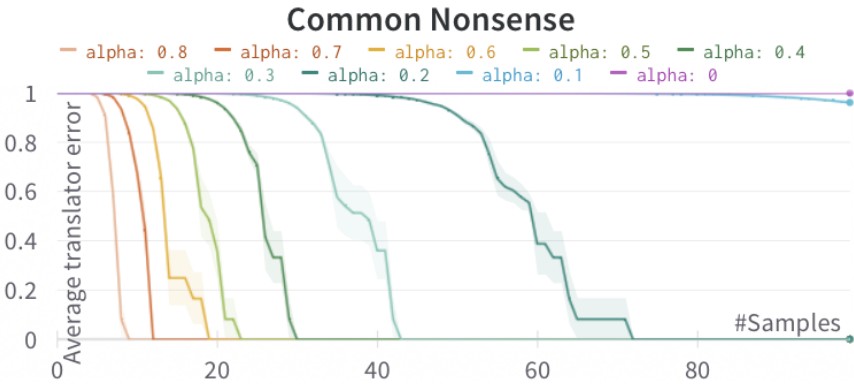

Figure 5: Common Nonsense model. The X-axis is the number of source samples $m$, and the Y-axis is the average error among plausible translators (that have not been ruled-out so far). Each experiment was run on five seeds, with standard error depicted by the shaded area.

**Contributions.** The first contribution of this work is formalizing and analyzing a model of UMT. As an initial work, its value is in the opportunities which it opens for further work more than the finality and tightness/generality of its bounds. Our model applies even to low-resource source languages with massive domain gap and linguistic distance. We emphasize that this work is only *a first step* in the theoretical analysis of UMT (indeed, there is little theoretical work on machine translation in general). Second, we exhibit two simple complementary models for which we prove that: (a) more complex languages require less common ground, and (b) data requirements may not be significantly greater than those of supervised translation (which tends to use less data than training a large LM). These findings may have implications for the quantity and type of communication data that is collected for deciphering animal communication and for UMT more generally. They also give theoretical evidence that UMT can be successful and worth pursuing, in lieu of parallel (supervised) data, in the case of sufficiently complex languages. All of that said, we note that our sample complexity bounds are information theoretic, that is, they do not account for the computational complexity of optimizing the translator. Finally, animal communication aside, to the best of our knowledge this work is the first theoretical treatment of UMT, and may also shed light on translation between human languages.

**Organization.** The framework is formally described in Section 2, and is instantiated with models of language in Section 3 (proofs, experiments, and other details deferred to Appendices A to E). Key takeaways from this work are presented in Section 4. Related and future work is discussed in Appendices F and G. We illustrate how prompting LMs may give priors in Appendix H. Appendix I sketches

a generalization of our framework to the settings of lossy translation and infinite translator families. Lastly, Appendix J proves sample complexity bounds for the settings of supervised translation.

## 2  The General Framework

We use $f\colon \mathcal{X} \hookrightarrow \mathcal{Y}$ to denote a 1–1 function, in which $f(x) \neq f(x')$ for all $x \neq x'$. For $S \subseteq \mathcal{X}$, we write $f(S) \coloneqq \{f(x) \mid x \in S\}$. The indicator $\mathbf{1}_P$ is 1 if the predicate $P$ holds, and 0 otherwise. The uniform distribution over a set $S$ is denoted by $\mathcal{U}(S)$, and $\log = \log_2$ denotes base-2 logarithm.

**Language and Prior.** A *source language* is a distribution $\mu$ over a set of possible texts $\mathcal{X}$. Similarly, a *target language* is a distribution $\nu$ over a set of possible texts $\mathcal{Y}$. When clear from context, we associate each language with its corresponding set of possible texts. A *prior* distribution $\rho$ over translations $\mathcal{Y}$ aims to predict the probability of observing each translation. One could naively take $\rho = \nu$, but Appendix H describes how better priors can focus on the domain of interest. Intuitively, $\rho(y)$ measures how "plausible" a translation $y$ is. For simplicity, we assume that $\mathcal{X}, \mathcal{Y} \subseteq \{0,1\}^*$ are finite, non-empty sets of binary strings. Appendix I.2 discusses extensions to infinite sets.

**Translators.** A *translator* is a mapping $f\colon \mathcal{X} \hookrightarrow \mathcal{Y}$. There is a known set of 1–1 functions $\{f_\theta\colon \mathcal{X} \hookrightarrow \mathcal{Y} \mid \theta \in \Theta\}$ with *parameter* set $\Theta$. Since parameters are assumed to be known and fixed, we will omit them from the theorem statements and algorithm inputs, for ease of presentation. Like $\mathcal{X}, \mathcal{Y}$, the set $\Theta$ is assumed to be finite. Appendix I.1 considers translators that are not 1–1.

**Divergence.** A translator $f_\theta$ and a distribution $\mu$ induce a distribution over $y = f_\theta(x)$, which we denote by $f_\theta \circ \mu$. The *divergence* between this distribution and $\rho$ is quantified using the Kullback–Leibler (KL) divergence,

$$\mathrm{D}(\theta) \coloneqq \mathrm{KL}(f_\theta \circ \mu \parallel \rho) = \mathop{\mathbb{E}}_{x \sim \mu} \left[ \log \frac{\mu(x)}{\rho\big(f_\theta(x)\big)} \right] = \sum_x \mu(x) \log \frac{\mu(x)}{\rho\big(f_\theta(x)\big)} \geq 0.$$

Note that since $\mathrm{D}(\theta) = \mathbb{E}_{x \sim \mu}\left[ -\frac{1}{m} \sum \log \rho(f_\theta(x_i)) \right] - \mathrm{H}(\mu)$, and $\mathrm{H}(\mu)$ is a constant independent of $\theta$, the MLE of Definition 1.1 approximately minimizes divergence.

**Ground truth.** In order to define semantic loss, we consider a *ground-truth translator* $f_\star$ for some $\star \in \Theta$. We can then define the (ground-truth) *translated language* $\tau = f_\star \circ \mu$ over $\mathcal{Y}$, obtained by taking $f_\star(x)$ for $x \sim \mu$. This is similar to the standard realizability assumption, and some of our bounds resemble Occam bounds with training labels $y_i = f_\star(x_i)$. Of course, the ground-truth translator $\star$ is *not* known to the unsupervised learning algorithm. In our setting, we further require that ground-truth translations never have 0 probability under $\rho$:

**Definition 2.1** (Realizable prior). $\mathrm{Pr}_{x \sim \mu}[\rho(f_\star(x)) = 0] = 0$, *or equivalently* $\mathrm{D}(\star) < \infty$.

**Semantic loss.** The semantic loss of a translator is defined with respect to a *semantic difference* function $\ell\colon \mathcal{Y} \times \mathcal{Y} \to [0, 1]$. This function, unknown to the learner, measures the difference between two texts from the target language $\mathcal{Y}$, with $\ell(y, y) = 0$ for all $y$. For a given semantic difference $\ell$ and ground-truth translator $f_\star$, we define the *semantic loss* of a translator $f_\theta$ by

$$\mathcal{L}(\theta) \coloneqq \mathop{\mathbb{E}}_{x \sim \mu} \big[ \ell(f_\star(x), f_\theta(x)) \big].$$

Of particular interest to us is the *semantic error* $\mathrm{err}(\cdot, \cdot)$, obtained when $\ell$ is taken to be the 0-1 difference $\ell_{01} = (y, y') = 1$ for all $y' \neq y$. Note that since any semantic difference $\ell$ is upper bounded by 1, the semantic error upper-bounds any other semantic loss $\mathcal{L}$. That is,

$$\mathcal{L}(\theta) \leq \mathrm{err}(\theta) \coloneqq \mathop{\mathrm{Pr}}_{x \sim \mu} [f_\theta(x) \neq f_\star(x)].$$

Section 3 analyzes this error $\mathrm{err}(\theta)$, which thus directly implies bounds on $\mathcal{L}(\theta)$.

# 3 Models of Language: Instantiating the General Framework

## 3.1 Random knowledge graphs

In this section, we define a model in which each text represents an edge between a pair of nodes in a knowledge graph. Both languages have knowledge graphs, with the source language weakly agreeing with an unknown subgraph of the target language.

We fix $\mathcal{X} = X \times X = X^2$ and $\mathcal{Y} = Y \times Y = Y^2$ with $r := |X|$ and $n := |Y|$. The set of translators considered is all mappings from the $r$ source nodes to the $n$ target nodes, namely $\Theta_{XY} = \{\theta : X \hookrightarrow Y\}$ and $f_\theta((u,v)) := (\theta(u), \theta(v))$. The random knowledge graph is parametrized by the number of source nodes $r$, target node set $\mathcal{Y}$, an edge density parameter $p \in (0,1)$ representing the expected fraction of edges present in each graph, and an agreement parameter $\alpha \in (0,1]$ representing the correlation between these edges. In particular, $\alpha = 1$ corresponds to the case where both graphs agree on all edges, and $\alpha = 0$ corresponds to the case where edges in the graphs are completely independent. These parameters are unknown to the learner, who only knows $X$ and $Y$ (and thus $\mathcal{X} = X^2, \mathcal{Y} = Y^2$).

**Definition 3.1** (Random knowledge graph). *For a natural number $r \leq |Y|$, the* $\mathrm{KG} = \mathrm{KG}(Y, r, p, \alpha)$ *model determines a distribution over sets $T, P \subseteq \mathcal{Y}$ (which determine distributions $\rho$ and $\mu$). The sets $T$ and $P$ are sampled as follows:*

1. *Set $P \subseteq \mathcal{Y}$ is chosen by including each edge $y \in \mathcal{Y}$ with probability $p$, independently.*

2. *Set $S \subseteq Y$ of size $|S| = r$ is chosen uniformly at random.*

3. *Set $T \subseteq S^2$ is chosen as follows. For each edge $y \in S^2$, independently,*

    (a) *With probability $\alpha$, $y \in T$ if and only if $y \in P$.*
    (b) *With probability $1 - \alpha$, toss another $p$-biased coin and add $y$ to $T$ if it lands on "heads"; that is, $y \in T$ with probability $p$, independently.*

It is easy to see that $T \subseteq S^2$ and $P \subseteq Y^2$ marginally represent the edges of Erdős–Rényi random graphs $G_{r,p}$ and $G_{n,p}$, respectively. Moreover, the event that $y \in T$ is positively correlated with $y \in P$: for each $y \in S^2$, since with probability $\alpha > 0$ they are identical and otherwise they are independent. Formally, the equations below describe the probability of $y \in T$ for each $y \in S^2$ after we fix $S$ and choosing $T \subseteq S^2$. Letting $q := (1 - p)$, for each $y \in S^2$:

$$\Pr[y \in T] = \Pr[y \in P] = p \tag{1}$$
$$\Pr[y \in T \setminus P] = \Pr[y \in P \setminus T] = (1 - \alpha)pq \tag{2}$$
$$\Pr[y \notin P \mid y \in T] = \Pr[y \notin T \mid y \in P] = \frac{(1 - \alpha)pq}{p} = (1 - \alpha)q \tag{3}$$

The last equality, shows that the probability of excluding a random $y \in T$ from $P$ is smaller than the probability of excluding a random "incorrect translation" $y' \neq y$, $\Pr[y' \notin P] = q > (1 - \alpha)q$.

We now describe how $\rho, \tau$, are determined from $T, P$ and how $\mu, \star$ may be chosen to complete the model description. The ground-truth target translated distribution $\tau := \mathcal{U}(T)$ is uniform over $T$. The prior $\rho$ is uniform over $P$, and then "smoothed" over the rest of the domain $\mathcal{Y}$. Formally,

$$\rho(y) := \begin{cases} \frac{1}{2} \cdot \left( \frac{1}{|P|} + \frac{1}{|\mathcal{Y}|} \right) & \text{if } y \in P \\ \frac{1}{2|\mathcal{Y}|} & \text{if } y \notin P. \end{cases}$$

The ground-truth translator $\star \in \Theta$ is obtained by sampling a uniformly random $\star : X \hookrightarrow S$. Lastly, we take $\mu = \mathcal{U}(f_\star^{-1}(T))$, which agrees with the definition of $\tau$.[8]

Next, we state the main theorem for this model, formalizing Theorem 1.2 from the introduction.

---

[8]Formally, the KG model outputs $T, P$ which may not determine $S$ if some nodes have zero edges. In that case, we choose $\star$ randomly among $\theta$ such that $f_\theta(\mathcal{X}) \supseteq T$. In the exponentially unlikely event that either $S$ or $T$ is empty, we define both $\tau, \rho$ to be the singleton distribution concentrated on $(y, y)$ for the lexicographically smallest $y \in Y$ and $\mu$ to concentrated on $(x, x)$ for $x = f_\star^{-1}(y)$. It is not difficult to see that MLE selects a translator with 0 error.

**Theorem 3.2** (Translatability in the KG model). *Fix any $m \geq 1$, $\emptyset \neq S \subseteq Y, \delta, \alpha, p \in (0,1)$, and let $r := |S|, n := |Y|, q = 1 - p$. Then, with probability $\geq 1 - \delta$ over $T, P$ from $\mathrm{KG}(S, Y, p, \alpha)$,*

$$\mathrm{err}(\hat{\theta}) \leq \max\left(\frac{64}{\alpha^2 pq^2 r^2} \ln \frac{6n^r}{\delta}, \frac{2}{\alpha q}\sqrt{\frac{2}{m} \ln \frac{6n^r}{\delta}}\right),$$

*where $\hat{\theta} = \mathrm{MLE}^\rho(x_1, x_2, \ldots, x_m)$ is from Definition 1.1. Simply, for $p < 0.99$, with probability $\geq 0.99$,*

$$\mathrm{err}(\theta) = O\left(\frac{\log n}{\alpha^2 pr} + \frac{1}{\alpha}\sqrt{\frac{r \log n}{m}}\right).$$

The proof, given in Appendix D, requires generalizing our theory to priors that have full support. Experimental validation of the theorem is described in Appendix E.

## 3.2 Common nonsense model

We next perform a "smoothed analysis" of arbitrary LMs $\mu, \rho$ that are uniform over sets that share a small amount of randomness, i.e., a small common random set has been removed from both. This shared randomness captures the fact that some texts are implausible in both languages and that this set has some complex structure determined by the laws of nature, which we model as random.

The $\alpha$-common-nonsense distribution is a meta-distribution over pairs $(\mu, \rho)$ which themselves are uniform distributions over perturbed versions of $P, T$. This is inspired by Smoothed Analysis [36]. Recall that $\mathcal{U}(S)$ denotes the uniform distribution over the set $S$.

**Definition 3.3** (Common nonsense). *The $\alpha$-common-nonsense distribution $\mathcal{D}_\alpha^{P,T}$ with respect to nonempty sets $T \subseteq P \subseteq \mathcal{Y}$ is the distribution over $\left(\rho = \mathcal{U}(P \cap S), \tau = \mathcal{U}(T \cap S)\right)$ where $S \subseteq \mathcal{Y}$ is formed by removing each $y \in \mathcal{Y}$ with probability $\alpha$, independently.[9]*

To make this concrete in terms of a distribution $\mu$ on $\mathcal{X}$, for any ground-truth translator $f_\star : \mathcal{X} \hookrightarrow \mathcal{Y}$, we similarly define a distribution $\mathcal{D}_{\alpha,\star}^{P,T}$ over $(\mu, \rho)$ where $\mu := \mathcal{U}(f_\star^{-1}(T \cap S))$ is the uniform distribution over the subset of $\mathcal{X}$ that translates into $\tau$. We now state the formal version of Theorem 1.3.

**Theorem 3.4** (Translatability in the CN model). *Let $\{f_\theta : \mathcal{X} \hookrightarrow \mathcal{Y} \mid \theta \in \Theta\}$ a family of translators, $\star \in \Theta$, $\alpha, \delta \in (0, 1/2]$, $T \subseteq P \subseteq \mathcal{Y}$, and $m \geq 1$. Then with probability $\geq 1 - \delta$, MLE run on $\rho$ and $m \geq 1$ iid samples from $\mu$ outputs $\hat{\theta}$ with,*

$$\mathrm{err}(\hat{\theta}) \leq \frac{6}{\alpha} \max\left(\frac{1}{m}, \frac{16}{|T|}\right) \cdot \ln \frac{6|\Theta|}{\delta}.$$

*Note that the probability is over both $(\mu, \rho)$ drawn from $\mathcal{D}_{\alpha,\star}^{P,T}$, and the $m$ iid samples from $\mu$. More simply, with probability $\geq 0.99$,*

$$\mathrm{err}(\hat{\theta}) = O\left(\frac{\log |\Theta|}{\alpha \min(m, |T|)}\right).$$

When the amount of shared randomness $\alpha$ is a constant, then this decreases asymptotically like the bound of supervised translation (Theorem J.1) up until a constant, similar to Theorem 3.2. For very large $m$, each extra bit describing the translator (increase by 1 in $\log |\Theta|$) amounts to a constant number of mistranslated $x$'s out of all $\mathcal{X}$. The proof is deferred to Appendix C.

We also prove the following lower-bound that is off by a constant factor of the upper bound.

**Theorem 3.5** (CN lower-bound). *There exists constants $c_1, c_2 \geq 1$ such that: for any set $T \subseteq \mathcal{Y}$, for any $m \geq 1$, any $\alpha \in (0, 1/2]$, and any $\Theta$ with $c_1 \leq \log |\Theta| \leq \alpha \min(m, |T|)$, there exists $\Theta'$ of size*

---

[9]Again, in the exponentially unlikely event that either $P \cap S$ or $T \cap S$ is empty, we define both $\tau, \rho$ to be the singleton distribution concentrated on the lexicographically smallest element of $\mathcal{Y}$, so MLE outputs a 0-error translator.

$|\Theta'| \le |\Theta|$ *such that, for any* $P \supseteq T$ *and any algorithm* $A^\rho : \mathcal{X}^m \to \Theta'$, *with probability* $\ge 0.99$ *over* $\star \sim \mathcal{U}(\Theta')$ *and* $(\mu, \rho)$ *drawn from* $\mathcal{D}_{\alpha,\star}^{P,T}$ *and* $x_1, \ldots, x_m \sim \mu$,

$$\mathrm{err}(\hat{\theta}) \ge \frac{\log |\Theta|}{c_2 \alpha \min(m, |T|)},$$

*where* $\hat{\theta} = A^\rho(x_1, x_2, \ldots, x_m)$.

The only purpose of $\Theta$ in the above theorem is to upper-bound the description length of translators, as we replace it with an entirely different (possibly *smaller*) translator family $\Theta'$ that still has the lower bound using $\log |\Theta| \ge \log |\Theta'|$. Since $\mathcal{U}(\Theta')$ is the uniform distribution over $\Theta'$, the ground-truth classifier is uniformly random from $\Theta'$. A requirement of the form $\log |\Theta| = O(\alpha \min(m, |T|))$ is inherent as otherwise one would have an impossible right-hand side error lower-bound greater than 1, though the constants could be improved.

The proof of this theorem is given in Appendix C.2, and creates a model with $O(\log n)$ independent "ambiguities" that cannot be resolved, with high probability over $S, x_1, x_2, \ldots, x_m$. Experimental validation of the theorem is described in Appendix E.

## 4 Discussion

We have given a framework for unsupervised translation and instantiated it in two stylized models. Roughly speaking, in both models, the error rate is inversely related to the amount of samples, common ground, and the language complexity. The first two relations are intuitive, while the last is perhaps more surprising. All error bounds were *information-theoretic*, meaning that they guarantee a learnable accurate translator, but learning this translator might be computationally intensive.

In both models, the translators are *restricted*. In the knowledge graph, the translators must operate node-by-node following an assumed compositional language structure.[10] In the common nonsense model, the restriction is based on the translator description bit length $\log |\Theta|$. To illustrate how such restrictions can be helpful, consider *block-by-block* translators which operate on limited contexts (e.g., by paragraph). Consider again the hypothetical example of Figure 1. Suppose the three texts are outputs of three translators $\Theta = \{A, B, C\}$. Let us suppose that translator A always produces accurate and natural translations, and further that all translators work paragraph-by-paragraph, as modern translation algorithms operate within some limited context window. In fact, one can imagine the translators of different paragraphs as a set of *isolated adversaries* where each adversary is trying to mistranslate a paragraph, knowing the ground-truth translation of their paragraph, while attempting to maintain the plausibility of the entire translation. If only the first-paragraph adversary mistranslates *reef* to *ocean basin*, then the translation lacks coherence and is unlikely. If the adversaries are in cahoots and coordinate to all translate *reef* to *ocean basin*, they would generate: *Have you seen mom? I just returned from the ocean basin. At the basin, there were a lot of sea turtles.* which has low probability $\approx 10^{-25}$, presumably because encoded in GPT-3's training data is the knowledge that there are no turtles deep in the ocean near the basin. While the adversary could also decide to change the word *turtle* to something else when it appears near *basin*, eventually it would get caught in its "web of deceit." The intuition is that, across sufficiently many translations, the prior will not "rule out" the ground-truth translations while very incorrect translators will be ruled out.

**Judging success.** Our analysis sheds some light on whether it is even possible to tell if translation without parallel data (UMT) is successful. A positive sign would be if millions of translations are fluent English accounts that are consistent over time across translations. In principle, however, this is what LM likelihood should measure (excluding consistencies across translations which sufficiently powerful LMs may be able to measure better than humans). We also considered a statistical distance (KL divergence) between the translations $f_{\hat{\theta}}(x)$ for $x \sim \mu$ and the prior $y \sim \rho$, and $\mu$ could be estimated given enough samples. If this distance is close to zero, then one can have predictive accuracy regardless of whether the translations are correct. This raises a related philosophical quandary: a situation in which two beings are communicating via an erroneous translator, but both judge the conversation to be natural.

---

[10]That is, we assume that each translator has a latent map from nodes in the source graph into nodes in the target graph, and edges are mapped from the source to target graphs in the natural way. The study of compositional communication systems, among humans and animals, has played a central role in linguistics [39].

## Acknowledgments and Disclosure of Funding

We thank Madhu Sudan, Yonatan Belinkov and the entire Project CETI team, especially Pratyusha Sharma, Jacob Andreas, Gašper Beguš, Michael Bronstein, and Dan Tchernov for illuminating discussions. This study was funded by Project CETI via grants from Dalio Philanthropies and Ocean X; Sea Grape Foundation; Rosamund Zander/Hansjorg Wyss, Chris Anderson/Jacqueline Novogratz through The Audacious Project: a collaborative funding initiative housed at TED.

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

## A    Translator Revisions and Plausible Ambiguities

Even though $\star$ is unknown, it will be convenient to define, for each $\theta \in \Theta$, a *revision* of $f_\star$ which is the permutation $\pi_\theta^\star \colon \mathcal{Y} \hookrightarrow \mathcal{Y}$ between $f_\theta$ and $f_\star$, $\pi_\theta^\star(y) := f_\theta(f_\star^{-1}(y))$.[11] We write $\pi_\theta = \pi_\theta^\star$ when $\star$ is clear from context, and $\pi_\star(y) \equiv y$ is the identity revision.

Note that the divergence $\mathrm{D}(\theta)$ and semantic loss $\mathcal{L}(\theta)$ can equivalently be defined using revisions,

$$\mathrm{D}(\theta) := \mathrm{KL}(f_\theta \circ \mu \parallel \rho) = \mathrm{KL}(\tau \parallel \pi_\theta^{-1} \circ \rho) = \underset{y \sim \tau}{\mathbb{E}} \Big[ \log \frac{\tau(y)}{\rho(\pi_\theta(y))} \Big],$$

$$\mathcal{L}(\theta) := \underset{x \sim \mu}{\mathbb{E}} \big[ \ell(f_\star(x), f_\theta(x)) \big] = \underset{y \sim \tau}{\mathbb{E}} \big[ \ell(y, \pi_\theta(y)) \big].$$

To relate divergence and loss, it is helpful to define a notion of *plausible ambiguities* which are $\theta$ whose revisions $\pi_\theta(y \sim \tau)$ are not too unlikely under the prior. These may also be of independent interest as they capture certain types of ambiguities that can only be resolved with supervision.

**Definition A.1** ($\gamma$-Plausible ambiguities)**.** *For any $\gamma \in [0, 1]$, $\star \in \Theta$, and distributions $\tau, \rho$ over $\mathcal{Y}$, the set of $\gamma$-plausible ambiguities $\mathcal{A}_\gamma = \mathcal{A}_\gamma^{\star; \tau, \rho} \subseteq \Theta$ is:*

$$\mathcal{A}_\gamma := \left\{ \theta \in \Theta \;\middle|\; \Pr_{y \sim \tau} [\rho(\pi_\theta(y)) = 0] \leq \gamma \right\}, \; \text{and } \varepsilon_\gamma := \max_{\theta \in \mathcal{A}_\gamma} \mathcal{L}(\theta).$$

For example, *left↔right* would constitute a $\gamma$-plausible ambiguity if one could swap the two words, making a $\leq \gamma$ fraction of the translations have 0 probability. Such a revision would have low loss if such swaps are considered similar in meaning. Condition 2.1 is equivalent to $\star \in \mathcal{A}_0$.

The quantity of interest is $\varepsilon_\gamma$, the maximum loss of any $\gamma$-plausible ambiguity.[12] Next, we prove that the loss of the translator output by the Maximum Likelihood Estimator is not greater than $\varepsilon_\gamma$ given $m \geq \frac{1}{\gamma} \ln \frac{|\Theta|}{\delta}$ examples.

**Theorem A.2.** *Let $\mu, \rho$ be probability distributions over $\mathcal{X}, \mathcal{Y}$, resp., and $f_\star$ satisfy Condition 2.1: $\Pr_\mu[\rho(f_\star(x)) = 0] = 0$. Fix any $\delta \in (0, 1)$, $m \geq 1$, and let $\gamma \geq \frac{1}{m} \ln \frac{|\Theta|}{\delta}$. Then,*

$$\Pr_{x_1, \dots, x_m \sim \mu} \left[ \mathcal{L}(\hat{\theta}) \leq \varepsilon_\gamma \right] > 1 - \delta,$$

*where $\hat{\theta} = \mathrm{MLE}^\rho(x_1, \dots, x_m)$ and $\varepsilon_\gamma$ is from Definition A.1.*

Since $\varepsilon_\gamma$ is non-decreasing in $\gamma$, as the number of examples increases the loss bound $\varepsilon_\gamma$ decreases to approach $\varepsilon_0$. This bound and its proof in Appendix A are analogous to the realizable Occam bound of supervised classification, see Appendix J.

*Proof of Theorem A.2.* MLE minimizes $\hat{v}(\theta) := \frac{1}{m} \sum_{i \leq m} \log \frac{1}{\rho(\pi_\theta(y_i))}$ over $\theta \in \Theta$, where $y_i = f_\theta(x_i)$. By the realizable prior assumption $\hat{v}(\star) < \infty$. Thus, for the algorithm to fail, there must be a "bad" $\hat{\theta} \in B := \{\theta \in \Theta \mid \mathcal{L}(\theta) > \varepsilon_\gamma\}$ with $\hat{v}(\theta) \leq \hat{v}(\star) < \infty$. If $\rho(\pi_\theta(y_i)) = 0$ for any $i$, then $\hat{v}(\pi_\theta) = \infty$. Note that by Definition A.1, $B \cap \mathcal{A}_\gamma = \emptyset$, thus for all $\theta \in B$: $\Pr[\rho(\pi_\theta(y)) = 0] > \gamma$ and,

$$\Pr \left[ \hat{v}(\theta) < \infty \right] < (1 - \gamma)^m \leq e^{-\gamma m} \leq \frac{\delta}{|\Theta|}.$$

By the union bound over $\theta \in B$, $\Pr[\exists \theta \in B \; \hat{v}(\theta) < \infty] < \delta$ since $|B| \leq |\Theta|$. $\qquad \square$

## B    The Tree-based Model

### B.1    A tree-based probabilistic model of language

The last model, which we view as secondary to the previous two, is a simpler version of the common nonsense model. It can be viewed as a "warm-up" leading up to that more general model, and we

---

[11]Formally, $f_\theta \circ f_\star^{-1} \colon f_\star(\mathcal{X}) \hookrightarrow \mathcal{Y}$ is only defined on $f_\star(\mathcal{X})$ but can be extended to a full permutation on $\mathcal{Y}$ in many ways. For concreteness, we take the lexicographically smallest extension on $\mathcal{Y} \subseteq \{0, 1\}^*$.

[12]For intuition, note that $\varepsilon_\gamma$ can be bounded using $\gamma$: $\frac{\gamma}{\varepsilon_\gamma} \geq \min_{\theta \in \Theta} \Pr_{y \sim \tau} \big[ \rho(\pi_\theta(y)) = 0 \mid \pi_\theta(y) \neq y \big]$.

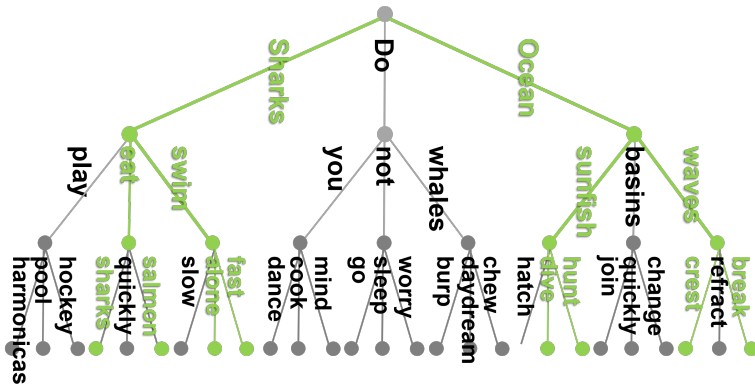

Figure 6: An example of a language tree of plausible texts and the subtree of ground-truth translations illustrated in green.

hope it helps illuminate that model by instantiating the language therein with a simple tree-based syntax.

In the tree-based model, the nodes of a tree are labeled with random words, and plausible texts (according to the prior $\rho$) correspond to paths from root to leaf. A translated language $\tau$ (or equivalently, a source language $\mu$) is then derived from root-to-leaf paths in a random subtree $H \subseteq G$. We prove that, with high probability, a prior $\rho$ and translated language $\tau$ sampled by this process satisfy Condition 2.1 and semantic error $\varepsilon_\gamma = O\big(1/\min(m, a^n)\big)$. Therefore, MLE yields a semantically accurate translator for the sampled language with high probability, for sufficiently large $n$.

The *random tree language* (RT) model is a randomized process for generating a tree-based source language $\mu$, translated language $\tau$, and prior $\rho$. It is parameterized by a finite vocabulary set $\mathcal{W}$, depth $n \in \mathbb{N}$, and arities $a, b \in \mathbb{N}$ such that $1 \le a \le b \le |\mathcal{W}|/4$. For simplicity, the possible target texts are taken to be the set of all possible $n$-grams over $\mathcal{W}$, namely, $\mathcal{Y} := \mathcal{W}^n$. Again for simplicity, we also assume $|\mathcal{X}| = |\mathcal{Y}|$ so that each $f_\theta \colon \mathcal{X} \hookrightarrow \mathcal{Y}$ is a bijection. To generate an RT, first a full $b$-ary, depth $n + 1$ tree $G$ is constructed. Labeled edges shall correspond to words, and paths shall correspond to texts (sentences), as follows:

1. Starting from the root node and proceeding in level-order traversal, for each node $v$: sample $b$ words $w_1, \ldots, w_b \in \mathcal{W}$ uniformly at random and *without* replacement; label each of $v$'s child edges with one of the sampled words.

2. The labels on each path from the root to a leaf $(y_1, \ldots, y_n)$ corresponds to a *plausible text*, giving a set of *plausible texts*

$$P := \{y \mid y = (y_1, \ldots, y_n) \text{ labels of a path in } G\}.$$

3. A subtree $H \subseteq G$ is obtained by sampling uniformly at random $a$ out of $b$ of the children at each level of the tree, in level-order traversal. The set of *translated texts* is analogously defined as

$$T := \{y \mid y = (y_1, \ldots, y_n) \text{ labels of a path in } H\} \subseteq P.$$

The prior $\rho = \mathcal{U}(P)$ is uniform over plausible texts $P$, while $\mu = $ is uniform over $f_\star^{-1}(T)$. We let $(\mu, \rho) \sim \mathrm{RT}(\mathcal{W}, n, a, b)$ denote the sampling of a prior and translated language obtained via this randomized process. When the parameters $\mathcal{W}, n, a, b$ are clear from context we simply write RT.

Note that $\Theta$ and $f_\star \in \Theta$ may be arbitrary, and by definition of $\tau$, $\mu = \mathcal{U}(f_\star^{-1}(T))$ is uniform over $T$'s preimage. Since we assumed $|\mathcal{X}| = |\mathcal{Y}|$ above, $f_\star$ is invertible.

Next, we will argue that a random tree language sampled in this process satisfies realizability (Condition 2.1) and Definition A.1 with appropriate choice of parameters. While we make no assumptions on $\Theta$, it is important that the plausible texts $P$ are sampled *after* the possible permutations (ambiguities) $\Pi$ are specified. Otherwise, if an adversary were able to choose a permutation $\pi \colon \mathcal{Y} \to$

$\mathcal{Y}$ based on $P$, then they could arbitrarily permute $P$, resulting in a permutation $\pi$ with high expected loss but no change in likelihood according to the prior $\rho$—thereby violating Definition A.1.

You can think of Theorem B.1 as pointing to the required text length $n$ and an upper bound on the number of parameters $|\Theta|$ for which Condition 2.1 and Definition A.1 hold with high probability. As we know from Theorem A.2, and state clearly next, these in turn imply translatability of a random tree language.

**Theorem B.1** (Translatability in the RT model). *Fix any $m \geq 1$, $\delta \in (0,1)$ vocabulary set $\mathcal{W}$, and tree arities $a \leq b \leq |\mathcal{W}|/4$. Then, for any $\Theta$ and any $\star \in \Theta$, with probability at least $1 - \delta$ over $(\mu, \rho) \sim \mathrm{RT}(\mathcal{W}, n, a, b)$ and iid samples $x_1, x_2, \ldots, x_m \sim \mu$,*

$$\mathrm{err}(\hat{\theta}) \leq 16 \max\left(\frac{1}{m}, \frac{4}{a^n}\right) \cdot \ln \frac{6|\Theta|}{\delta},$$

*where $\hat{\theta} = \mathrm{MLE}^\rho(x_1, x_2, \ldots, x_m)$ and $\mathrm{MLE}$ is from Definition 1.1*

Note that the probability in the corollary is over both $(\mu, \rho) \sim \mathrm{RT}$ and the $m$ iid training samples. Again, note how the first term is similar to the $\frac{1}{m} \log \frac{|\Theta|}{\delta}$ term from realizable supervised learning (see Theorem J.1), and the second term is additional due to the unsupervised case. The proof, given in Appendix B, uses Theorem A.2 and a lemma stating that $\varepsilon_\gamma \leq 16\gamma$ for $\gamma$ that are not too small. The main challenge is that there are dependencies between the paths, so one cannot directly use standard concentration inequalities.

To better understand Theorem B.1, let us consider two possible families of translators and suppose we are trying to apply Theorem B.1 to get bounds on the sample complexity $m$ needed to translate $(\mu, \rho) \sim \mathrm{RT}(\mathcal{W}, n, a, b)$ with small constant loss and small constant failure probability.

First, since $|\mathcal{X}| = |\mathcal{Y}| = |\mathcal{W}|^n$, the above bound is meaningless (bigger than 1) if $\Theta$ is the family of all translators, as the set of all translators has $\log(|\mathcal{X}|!) = O(|\mathcal{W}|^n \cdot n \log |\mathcal{W}|)$ parameters which is much larger than $a^n$. In other words, *no free lunch*.

On the other hand, let us consider the family of *word-for-word* translators $\Theta_{\mathrm{w}}$, which work by translating each word in a text separately, ignoring the surrounding context. The number of such translators is the same as the number of word-to-word permutations, i.e., $|\Theta_{\mathrm{w}}| = |\mathcal{W}|!$. So Theorem B.1 gives a sample complexity bound of $m = O(\log \mathcal{W}!) = O(|\mathcal{W}| \log |\mathcal{W}|)$. The number of words in the English language is about $N \approx 10^5$. The quantity $a$ is equal to what is called the perplexity of $\mu$, the effective number of words that may typically appear after a random prefix. For a constant $a$, the minimum text (or communication) length $n = O(\log N)$ needed is thus logarithmic in the vocabulary size. While word-for-word translators are poor, this analysis still sheds some light on the possible asymptotic behavior (barring computational constraints) of translation, at least in a simplistic model.

**Common tree analysis.** The tree model helps us demonstrate the generality of the common nonsense model. Instead of the fixed arities $a \leq b$ in the tree model, one can consider an *arbitrary* language tree over plausible texts $P$ and an *arbitrary* subset $T \subseteq P$ corresponding to a subtree of ground-truth translations. The common nonsense model implies that if the sets are perturbed by removing, say, a $\alpha = 0.01$ common fraction of translations, then with high probability MLE's error decreases at an $\tilde{O}(1/\min(m, |T|))$ rate. Note that Theorem 3.4 does not directly apply to the random tree LM because in that LM, the choice of which branches to include are not independent due to the fixed $a$-arity constraint.

## B.2 Proof of Theorem B.1

The proof of Theorem B.1 follows from Lemma B.2 below. In this section, we will prove this lemma and then derive the theorem from it.

**Lemma B.2** (RT conditions). *Consider any vocabulary set $\mathcal{W}$, tree arities $a \leq b \leq |\mathcal{W}|/4$, tree depth $n \in \mathbb{N}$. Then, for any $\delta \in (0,1)$ and $\gamma \geq 4a^{-n} \ln(4|\Theta|/\delta)$, with probability $\geq 1 - \delta$, $(\mu, \rho)$ give,*

$$\Pr_{(\mu, \rho) \sim \mathrm{RT}} [\varepsilon_\gamma \leq 16\gamma] \geq 1 - \delta. \tag{4}$$

*Moreover, any $\mu, \rho$ sampled from $\mathrm{RT}(\mathcal{W}, n, a, b)$ satisfy $\Pr_{x \sim \mu}[\rho(f_\star(x)) = 0] = 0$ as needed for the realizability Condition 2.1.*

The proof of Lemma B.2 requires two additional lemmas, stated and proved next. The first is a known variant of the Chernoff bound for so-called 0-negatively correlated random variables.

**Lemma B.3** (Chernoff bound for 0-negatively correlated random variables)**.** *Let $Z_1, \ldots, Z_n$ be 0-negatively correlated Boolean random variables, that is, they satisfy*

$$\forall I \subseteq [n] \quad \Pr\left[\forall i \in I \ Z_i = 0\right] \leq \prod_{i \in I} \Pr\left[Z_i = 0\right].$$

*Then, letting $Z := \sum_{i=1}^{n} Z_i$ it holds that*

$$\Pr\left[Z \leq \frac{\mathbb{E}[Z]}{2}\right] \leq e^{-\frac{\mathbb{E}[Z]}{8}}$$

Lemma B.3 follows from [14, Theorem 1.10.24(a)] with $\delta := 1/2$, $a_i := 0$, $b_i := 1$ and $Y_i := X_i$. That theorem, in turn, is simply Theorem 1.10.10 for 0-negatively correlated random variables.

The second lemma used for the proof of Lemma B.2 is a combinatorial argument that we prove below.

**Lemma B.4.** *Given any $\pi : \mathcal{Y} \hookrightarrow \mathcal{Y}$, it is possible to partition $A = \{y \in \mathcal{Y} \mid \pi(y) \neq y\}$ into four disjoint sets $A = A_1 \cup A_2 \cup A_3 \cup A_4$ such that, for each $i = 1, 2, 3, 4$, the following two conditions hold:*

$$\pi(A_i) \cap A_i = \emptyset$$

$$\forall z \in \mathcal{W}^{n-1} \ |\{y \in A_i \mid y \text{ begins with } z\}| \leq \frac{|\mathcal{W}|}{2}$$

*Proof.* of Lemma B.4. We will partition $A$ greedily to achieve the two conditions. Begin with four empty sets $A_1 = A_2 = A_3 = A_4 = \emptyset$. For each $y \in A$ in turn, assign it to one of the 4 sets as follows.

1. Let $i$ be the index of the set $A_i$ such that $\pi(y) \in A_i$ has already been assigned to. If $\pi(y))$ has not yet been assigned, let $i = 1$.

2. Similarly, if $\pi(y) \in A_j$ has been assigned, let $j$ be its index otherwise $j = 1$.

3. Let $z$ be the first $n - 1$ elements of $x$. Let $k$ be the index of the set $A_k$ such that $|\{y \in A_i \mid y \text{ begins with } z\}| \geq |\mathcal{W}|/2$ if there is such a set, otherwise $k = 1$. Note that there can be at most one such $k$ because there are at most $|\mathcal{W}| - 1$ other elements beginning with $z$ that have been assigned already.

Thus $S = \{1, 2, 3, 4\} \setminus \{i, j, k\} \neq \emptyset$ and we can assign $x$ to any (say the minimum) element in that set. By induction, we preserve the two properties stated in the lemma. $\square$

With Lemmas B.3 and B.4 in hand, we are ready to prove Lemma B.2.

*Proof.* of Lemma B.2. The realizability Condition 2.1, $\Pr_{x \sim \mu}[\rho(f_\star(x)) = 0] = 0$, follows immediately from the fact that $\rho$ is uniform over $P$ and that $\tau$ is supported on $T \subseteq P$. To prove the lemma, it suffices to show that for any $\gamma \geq 4a^{-n} \ln(4|\Theta|/\delta)$,

$$\Pr_{(\mu, \rho) \sim \text{RT}}\left[\exists \theta \in \Theta \ \ \text{err}(\theta) > 16\gamma, \ \Pr_{y \sim \tau}[\rho(\pi_\theta(y)) = 0] \leq \gamma\right] \leq \delta.$$

Note that $\text{err}(\theta) = \Pr_{y \sim \tau}[y \neq \pi_\theta(y)]$, and that $\rho(\pi_\theta(y)) = 0$ if and only if $\pi_\theta(y) \notin P$. Therefore, by the union bound over $\theta \in \Theta$, it suffices to show that, for each $\theta \in \Theta$,

$$\Pr_{(\mu, \rho) \sim \text{RT}}\left[\Pr_{y \sim \tau}[y \neq \pi_\theta(y)] > 16\gamma, \ \Pr_{y \sim \tau}[\pi_\theta(y) \notin P] \leq \gamma\right] \leq \frac{\delta}{|\Theta|}. \tag{5}$$

We will show that, more generally, for any permutation $\pi : \mathcal{Y} \hookrightarrow \mathcal{Y}$. Equation (5) can be restated in terms of *ambiguous texts* $A := \{y \in \mathcal{Y} \mid \pi(y) \neq y\}$ and *implausible ambiguities* $B := \{y \in \mathcal{Y} \mid \pi(y) \notin P\}$ :

$$\Pr_{(\mu,\rho)\sim\text{RT}}[\tau(A) > 16\gamma,\ \tau(B) \le \gamma] < \frac{\delta}{|\Theta|}.$$

Using Lemma B.4, partition $A$ into four disjoint sets $A = A_1 \cup A_2 \cup A_3 \cup A_4$ such that for each $i \in [4]$, the following two conditions hold:

$$\pi(A_i) \cap A_i = \emptyset \tag{6}$$

$$\forall z \in \mathcal{W}^{n-1} \quad |\{y \in A_i \mid y \text{ begins with } z\}| \le \frac{|\mathcal{W}|}{2}. \tag{7}$$

By the union bound, it suffices to show that,

$$\forall i \in [4] \quad \Pr_{(\mu,\rho)\sim\text{RT}}[\tau(A_i) > 4\gamma,\ \tau(B) \le \gamma] < \frac{\delta}{4|\Theta|},$$

because $\tau(A) = \sum_{i=1}^4 \tau(A_i)$, so if $\tau(A) > 16\gamma$ then it must follow that $\tau(A_i) > 16\gamma/4$ for some $i$. It suffices to show that this holds conditioned on any value of $A_i \cap T$:

$$\forall V \subseteq A_i \quad \Pr_{(\mu,\rho)\sim\text{RT}}[\tau(A_i) > 4\gamma, \tau(B) \le \gamma \mid A_i \cap T = V] \le \frac{\delta}{4|\Theta|}. \tag{8}$$

Thus, fix any $V \subseteq A_i$. We proceed by analyzing two cases, based on $|V|$ versus $|T| = a^n$.[13]

**Case 1:** $|V| \le 4\gamma|T|$. Then Equation (8) holds with probability 0, because conditioning on $A_i \cap T = V$, we have

$$\tau(A_i) := \frac{|A_i \cap T|}{|T|} = \frac{|V|}{|T|} \le 4\gamma.$$

**Case 2:** $|V| > 4\gamma|T|$. For each $v \in V$ we define a Boolean random variable $Z_v = \mathbf{1}_{v \in B}$, i.e.,

$$Z_v = \begin{cases} 1 & v \in B \\ 0 & v \notin B. \end{cases}$$

We claim that the $Z_v$'s satisfy the condition of Lemma B.3, which follows inductively from the fact any subset of $Z_v$'s being 0 only makes it *less* likely for another $Z_v$ to be 0, because the way that the $Z_v$'s are chosen is such that there are exactly $b$ many 1's among the $Z_v$'s (and other symmetric random variables which we have not named).

Therefore, if we define $Z := \sum_{v \in V} Z_v$ and $\zeta := \mathbb{E}[Z \mid A_i \cap T = V]$, Lemma B.3 gives,

$$\Pr\left[Z \le \frac{\zeta}{2} \,\Big|\, A_i \cap T = V\right] \le e^{-\zeta/8}. \tag{9}$$

We conclude by bounding both sides of Equation (9) to obtain Equation (8).

- For the right-hand side of eq. (9), we claim that $\Pr[Z_v = 1 \mid A_i \cap T = V] \ge 1/2$ for each $v \in V$: Fix $v$ and let

$$Q = \{y \in \mathcal{Y} \mid y \text{ and } \pi(v) \text{ agree on the first } n-1 \text{ words}\}.$$

So $|Q| = |\mathcal{W}|$ and $|Q \cap P| = b$. Equation (6) implies that $\pi(v) \notin A_i$, and eq. (7) implies that $|Q \setminus A_i| \ge |\mathcal{W}|/2$. Thus, there are $\ge |\mathcal{W}|/2$ elements in $Q \setminus A_i$ that, by symmetry, are as likely to be in $Q \cap P$ as $\pi(v)$ is, and $Q \cap P$ contains exactly $b \le |\mathcal{W}|/4$ elements, thus the conditional probability that $\pi(v)$ is in $Q \cap P$ (equivalently, $\pi(v) \in P$) is at most:

$$\frac{|\mathcal{W}|/4}{|\mathcal{W}|/2} \le \frac{1}{2}.$$

Since $\pi(v) \notin P$ is equivalent to $Z_v = 1$, this means that $\Pr[Z_v = 1 \mid A_i \cap T = V] \ge 1/2$, and hence:

$$\zeta = |V| \cdot \mathbb{E}[Z_v \mid A_i \cap T = V] = |V| \cdot \Pr[Z_v = 1 \mid A_i \cap T = V] \ge \frac{|V|}{2}.$$

---

[13] Note that while $T$ is randomly sampled, $|T| = a^n$ always, and therefore the case-analysis is valid.

Because we are in the case that $|V| > 4\gamma|T|$, we have

$$\zeta \geq \frac{|V|}{2} > \frac{1}{2} \cdot 4\gamma \cdot |T| = 2\gamma \cdot a^n. \tag{10}$$

Therefore the right-hand side of eq. (9) satisfies

$$e^{-\zeta/8} \leq e^{-\gamma a^n/4} \geq \frac{\delta}{4\,|\Theta|},$$

where the rightmost inequality holds because by our assumption that $\gamma \geq 4a^{-n}\ln(4|\Theta|/\delta)$.

- For the left-hand side of eq. (9), note that

$$Z := \sum_{v \in V} Z_v := \sum_{v \in V} \mathbf{1}_{v \in B} = |V \cap B|.$$

and therefore, using eq. (10), the left-hand side of eq. (9) satisfies

$$\Pr\left[Z < \frac{\zeta}{2} \;\middle|\; A_i \cap T = V\right] \geq \Pr\left[\frac{|V \cap B|}{|T|} \leq \gamma \;\middle|\; A_i \cap T = V\right].$$

We claim that $V \cap B \subseteq B \cap T$: Due to the conditional event, we have $V \cap B = A_i \cap T \cap B \subseteq A \cap T \cap B$. However, we argue that $A \cap T \cap B \subseteq T \cap B$, i.e., that $T \cap B \subseteq A$. Indeed, by definition, if $y \in T \cap B$ then $y \in T$ but $\pi(y) \notin P$, and since $P \supseteq T$, this implies that $\pi(y) \neq y$; that is, that $y \in A$, as needed.

We have just shown that $V \cap B \subseteq B \cap T$, and therefore

$$\Pr\left[\frac{|V \cap B|}{|T|} \leq \gamma \;\middle|\; A_i \cap T = V\right] \geq \Pr\left[\frac{|B \cap T|}{|T|} \leq \gamma \;\middle|\; A_i \cap T = V\right]$$
$$= \Pr\left[\tau(B) \leq \gamma \mid A_i \cap T = V\right].$$

This shows that the left-hand side of eq. (9) upper bounds that of eq. (8), and concludes the proof.

$\square$

Lastly, we prove Theorem B.1 using Lemma B.2

*Proof.* of Theorem B.1. Let $\gamma$ be,

$$\gamma := \max\left(\frac{1}{m}, \frac{4}{a^n}\right)\ln\frac{6|\Theta|}{\delta}.$$

Lemma B.2 below shows that with probability $\geq 1 - 2\delta/3$ over $\mu, \rho$, we have that $\varepsilon_\gamma \leq 16\gamma$. Theorem A.2 implies that with probability $\geq 1 - \delta/3$ over $x_1, \ldots, x_m$, we have $\mathrm{err}(\hat{\theta}) \leq \varepsilon_\gamma$. Thus, by the union bound, with probability $\geq 1 - \delta$ we have

$$\mathrm{err}(\hat{\theta}) \leq \varepsilon_\gamma \leq 16\gamma = 16\max\left(\frac{1}{m}, \frac{4}{a^n}\right)\ln\frac{6|\Theta|}{\delta}.$$

$\square$

## C  Proofs for the Common Nonsense Model

### C.1  Upper bound in the common nonsense model

#### C.1.1  Proof overview

To convey intuition, we think of the case of say $\delta = 99\%$, and we omit constants in the following discussion. Recall that Theorem A.2 asserts that with high probability, the learned translator $\hat{\theta}$ has error at most $\varepsilon_\gamma$ for $\gamma = \log|\Theta|/\min(m, |T|) \geq \log|\Theta|/m$. As such, the main technical challenge

in this proof is to show that, with high probability over the generated source language $\mu$ and the prior $\rho$, it holds that

$$\varepsilon_\gamma \lesssim \frac{\gamma}{\alpha}.$$

By definition of $\varepsilon_\gamma$, we ought to show that w.h.p over $\mu$ and $\rho$, any $\theta \in \Theta$ with large semantic error must have many translations $f_\theta(x)$ deemed implausible by $\rho$. Slightly more formally: Since any $y \in \mathcal{Y}$ is implausible ($\rho(y) = 0$) only if $y \notin S$, a union bound over $\theta \in \Theta$ means that is suffices to show that

$$\Pr_{S,\mu,\rho}\left[ \Pr_{x\sim\mu}[f_\theta(x) \neq f_\star(x)] \lesssim \frac{\gamma}{\alpha}, \ \Pr_{x\sim\mu}[f_\theta(x) \notin S] \gtrsim \gamma \right] \lesssim \exp(-\gamma|T|)) \leq \frac{1}{|\Theta|},$$

where the right inequality is by choice of $\gamma := \log|\Theta| / \min(m, |T|)$. The above inequality "looks like" it could be proven by a Chernoff bound, but a closer look reveals a subtle flaw with this argument.

To use a Chernoff bound, we'd first want to fix (i.e., condition on) each $\mathrm{supp}(\mu) := f_\star^{-1}(S \cap T)$ and then use Chernoff over the conditional random variables $\mathbf{1}_{f_\theta(x) \notin S}$ for each $x \in \mathrm{supp}(\mu)$ such that $f_\theta(x) \neq f_\star(x)$. Unfortunately, these conditional random variables are *not* independent. To see this, consider the case that $f_\theta(x) = f_\star(x')$ for two different $x \neq x'$. Then, since we are considering $x' \in \mathrm{supp}(\mu)$, we have $f_\theta(x) = f_\star(x') \in S$ with probability 1.

To avoid this dependency, we prove a combinatorial lemma showing that it is possible to partition the set

$$A := \{x \in X \mid f_\theta(x) \neq f_\star(x)\}$$

into three parts $A = A_1 \cup A_2 \cup A_3$ such that $f_\theta(A_i) \cap f_\star(A_i) = \emptyset$ for each $i \in [3]$. This resolves the dependency issue demonstrated above. We then proceed by applying a Chernoff bound separately for each $A_i$, which suffices since a union bound (over $i \in [3]$) loses only a constant factor in the upper-bound.

### C.1.2 The full proof

The proof of Theorem 3.4 follows from the following main lemma. We first prove this lemma, and then show how the theorem follows from it.

**Lemma C.1.** *Let* $\alpha, \delta \in (0,1)$ *and* $T \subseteq P \subseteq \mathcal{Y}$. *Then, for any* $\gamma \geq \frac{8}{(1-\alpha)\cdot|T|} \ln \frac{4|\Theta|}{\delta}$:

$$\Pr_{(\mu,\rho)\sim\mathcal{D}_\alpha^{P,T}}\left[\varepsilon_\gamma \leq \frac{6\gamma}{\alpha}\right] \geq 1 - \delta.$$

The proof of Lemma C.1 relies on a simple combinatorial proposition. This proposition is a special case of Lemma B.4, but since it is much simpler we give a self-contained proof.

**Proposition C.2.** *For any finite set* $\mathcal{Y}$ *and any* $\pi : \mathcal{Y} \hookrightarrow \mathcal{Y}$, *it is possible to partition* $\{y \in \mathcal{Y}|\pi(y) \neq y\} = A_1 \cup A_2 \cup A_2$ *into three sets such that,* $\pi(A_i) \cap A_i = \emptyset$ *for* $i = 1, 2, 3$.

*Proof.* Let $S := \{y \in \mathcal{Y} \mid \pi(y) \neq y\}$. We proceed iteratively, dividing each (non-trivial) cycle of $\pi$ separately into the three $A_i$'s: Fix a cycle $\{s_1, s_2, \ldots, s_n\}$ such that $\pi(s_i) = s_{i+1}$ and $\pi(s_n) = s_1$. If $n$ is even we can just partition it into two sets: put the $s_i$ for even $i$'s into $A_1$ and odd $i$'s into $A_2$. If $n$ is odd, we can do the same except put the last element $s_n$ into $A_3$. $\square$

We can now prove Lemma C.1.

*Proof.* of Lemma C.1. Note that the lemma holds trivially for any $\gamma > 1/6$ because we always have $\varepsilon_\gamma \leq 1$. Assume that $\gamma \leq 1/6$. Let $a := \frac{6}{\alpha}\gamma$. The probabilities in this proof are over the choice of $S \subseteq \mathcal{Y}$. It suffices to show that,

$$\Pr_S\left[\exists\theta \in \Theta \ \Pr_\tau[\pi_\theta(y) \neq y] > a \ \wedge \ \Pr_\tau[\pi_\theta(y) \notin S] \leq \gamma\right] \leq \delta, \tag{11}$$

because $\rho(\pi_\theta(y)) = 0$ whenever $\pi_\theta(y) \notin S$, thus $\mathcal{A}_\gamma = \{\theta \in \Theta \mid \Pr_\tau[\pi_\theta(y) \notin S] \leq \gamma\}$. The set $S$ determines the perturbed sets $\tilde{T} := T \cap S$ and $\tilde{P} := P \cap S$ and the distributions $\tau = \mathcal{U}(\tilde{T})$ and $\rho = \mathcal{U}(\tilde{P})$.

Define $\beta := 1 - \alpha$. Since $\mathbb{E}\big[|\tilde{T}|\big] = \beta|T|$, a multiplicative Chernoff bound[14] gives

$$\Pr_S\left[|\tilde{T}| \le \frac{\beta}{2}|T|\right] \le \exp\left(-\frac{\beta}{8}|T|\right) \le \exp\left(-\frac{\beta}{8} \cdot \frac{8}{\beta\gamma} \ln \frac{4|\Theta|}{\delta}\right) < \frac{\delta}{4|\Theta|} \le \frac{\delta}{4}.$$

Thus, to show eq. (11), it suffices to show that

$$\Pr\left[|\tilde{T}| \ge \frac{\beta}{2}|T| \;\wedge\; \exists\theta \in \Theta \; \Pr_\tau[\pi_\theta(y) \ne y] > a \;\wedge\; \Pr_\tau[\pi_\theta(y) \notin S] \le \gamma\right] \le \frac{3\delta}{4}. \tag{12}$$

By the union bound, to show eq. (12) it thus suffices to show that for any $\pi : \mathcal{Y} \hookrightarrow \mathcal{Y}$,

$$\Pr\left[|\tilde{T}| \ge \frac{\beta}{2}|T| \;\wedge\; \Pr_\tau[\pi(y) \ne y] > a \;\wedge\; \Pr_\tau[\pi(y) \notin S] \le \gamma\right] \le \frac{3\delta}{4|\Theta|}. \tag{13}$$

Fix any $\pi : \mathcal{Y} \hookrightarrow \mathcal{Y}$. By Lemma C.2, we can partition $\{y \in \mathcal{Y} \mid \pi(y) \ne y\} = A_1 \cup A_2 \cup A_3$ such that $\pi(A_i) \cap A_i = \emptyset$, and hence $\Pr_\tau[\pi(y) \ne y] = \sum_i \tau(A_i)$. So if $\Pr_\tau[\pi(y) \ne y] > a$, then $\tau(A_i) > a/3$ for some $i$. Therefore, it suffices to show that

$$\Pr\left[|\tilde{T}| \ge \frac{\beta}{2}|T| \;\wedge\; \exists i \in [3] \; \tau(A_i) > \frac{a}{3} \;\wedge\; \Pr_\tau[\pi(y) \notin S] \le \gamma\right] \le \frac{3\delta}{4|\Theta|}.$$

With a union bound over $i \in [3]$, it suffices to show that for each $i \in [3]$

$$\Pr\left[|\tilde{T}| \ge \frac{\beta}{2}|T| \;\wedge\; \tau(A_i) > \frac{a}{3} \;\wedge\; \Pr_\tau[\pi(y) \notin S] \le \gamma\right] \le \frac{\delta}{4|\Theta|}. \tag{14}$$

Thus, now in addition to fixing $\pi$, we fix $i \le 3$, thus fixing $A_i$. To continue, imagine we are picking $S$ by first selecting $A_i \cap S$, and subsequently selecting $S \setminus A_i$. We will show that Equation (14) holds when conditioning on each possible value for the first selection, that is, each possible $A_i \cap S$. Formally, we fix $V \subseteq A_i$ and condition on $A_i \cap S = V$, claiming that

$$\Pr\left[|\tilde{T}| \ge \frac{\beta}{2}|T| \;\wedge\; \tau(A_i) > \frac{a}{3} \;\wedge\; \Pr_\tau[\pi(y) \notin S] \le \gamma \;\middle|\; V = A_i \cap S\right] \le \frac{\delta}{4|\Theta|}. \tag{15}$$

First, observe that

$$\tau(A_i) = \frac{|A_i \cap \tilde{T}|}{|\tilde{T}|} = \frac{|A_i \cap S|}{|\tilde{T}|} = \frac{|V|}{|\tilde{T}|},$$

therefore if $|V| < \beta a|T|/6$ then Equation (15) holds with probability 0 (due to the first two events in the conjunction). Thus, we can assume that $|V| \ge \beta a|T|/6$.

Note that $\tau(V) = \tau(A_i)$, and that $\tau(A_i) \ge a/3$ implies

$$\Pr_{y \sim \tau}[\pi(y) \notin S] \ge \Pr_{y \in V}[\pi(y) \notin S] \cdot \tau(V) \ge \Pr_{y \in V}[\pi(y) \notin S] \cdot \frac{a}{3},$$

therefore the left-hand side of Equation (15) is upper-bounded by

$$\Pr\left[\Pr_{y \in V}[\pi(y) \notin S] \le \frac{3 \cdot \gamma}{a} \;\middle|\; V = A_i \cap S\right] = \Pr\left[\Pr_{y \in V}[\pi(y) \notin S] \le \frac{a}{2} \;\middle|\; V = A_i \cap S\right]. \tag{16}$$

We conclude the proof by upper-bounding Equation (16) with a Chernoff bound. Consider the random variables $Z_y = \mathbf{1}_{\pi(y) \notin S}$ for $y \in V$. These random variables are independent by definition of $\alpha$-common-nonsense. Furthermore, due to the fact that $\pi(A_i) \cap A_i = \emptyset$, they remain independent even when conditioning on the event $A_i \cap S = V$. By linearity of expectation,

$$\mathbb{E}\left[\sum_{y \in V} Z_y \;\middle|\; V = A_i \cap S\right] = \alpha|V|.$$

---

[14]Specifically, that the probability that a sum of binary random variables is less than half its mean $\beta|T|$ is at most $\exp(-\beta|T|/8)$.

Using the same Chernoff bound as above, we have

$$\Pr\left[\frac{\sum_{y \in V} Z_y}{|V|} \leq \frac{\alpha}{2} \,\Big|\, V = A_i \cap S\right] \leq \exp(-\alpha|V|/8)$$

Noting that $\sum Z_y/|V| = \Pr_{y \in V}[\pi(y) \notin S]$, we conclude that that Equation (16) is upper-bounded by

$$\exp\left(-\frac{\alpha|V|}{8}\right) \leq \exp\left(-\frac{\alpha\beta a|T|}{48}\right) \leq \frac{\delta}{4|\Theta|}.$$

This proves the inequality in Equation (15), thereby concluding the proof.

$\square$

Finally, with the main technical lemma in hand, we prove Theorem 3.4.

*Proof.* of Theorem 3.4. The proof is very similar to that in Appendix B. First, the realizability Condition 2.1, $\Pr_{x \sim \mu}[\rho(f_\star(x)) = 0] = 0$, follows immediately from the fact that $\mu, \rho$ are uniform distributions with $f_\star(\text{supp}(\mu)) \subseteq \text{supp}(\rho)$ which follows from the fact that $T \subseteq P$ and the definitions of $\mu, \rho$.

Let $\beta := 1 - \alpha$ and define $\gamma$ by,

$$\gamma := \max\left(\frac{1}{m}, \frac{8}{\beta|T|}\right) \ln \frac{6|\Theta|}{\delta}.$$

Lemma C.1 below shows that with probability $\geq 1 - 2\delta/3$ over $\mu, \rho$, we have that $\varepsilon_\gamma \leq 6\gamma/\alpha$. Theorem A.2 implies that with probability $\geq 1 - \delta/3$ over $x_1, \ldots, x_m$, we have $\text{err}(\hat{\theta}) \leq \varepsilon_\gamma$. Thus, by the union bound, with probability $\geq 1 - \delta$ we have

$$\text{err}(\hat{\theta}) \leq \varepsilon_\gamma \leq \frac{6\gamma}{\alpha} = \frac{6}{\alpha} \max\left(\frac{1}{m}, \frac{8}{\beta|T|}\right) \ln \frac{6|\Theta|}{\delta}.$$

$\square$

### C.2 Lower bound in the common nonsense model

The proof of the lower bound works by creating $\log n$ candidate "plausible ambiguities" and arguing that a constant fraction of survive the random removal of elements.

*Proof.* of Theorem 3.5. The constants $c_1, c_2$ will also be determined through this proof to be large enough to satisfy multiple conditions defined below. No effort has been made to minimize the constants in this proof.

Let $n := |\Theta|$.

We will lay out two $a \times b$ grids $X \subseteq \mathcal{X}$ and $Y \subseteq \mathcal{Y}$, for:

$$a := \lfloor \log n \rfloor, b := \left\lfloor \frac{1}{\alpha} \max\left(1, \frac{|T|}{10^5 m}\right) \right\rfloor.$$

For integer $t$, denote $[t] := \{1, 2, \ldots, t\}$. For $i \in [a], j \in [b]$, choose distinct elements $x_{ij} \in \mathcal{X}$ and $y_{ij} \in T$. To ensure this is even possible, we must make sure $ab \leq |T|$, which holds because we assumed $\log n \leq \alpha \min(m, |T|)$ thus,

$$ab \leq \alpha \min(m, |T|) \frac{1}{\alpha} \max\left(1, \frac{|T|}{m}\right) = \min(m, |T|) \cdot \max\left(\frac{1}{|T|}, \frac{1}{m}\right) |T| = |T|.$$

Let $X := \{x_{ij} \mid i \in [a], j \in [b]\}$ and $Y := \{y_{ij} \mid i \in [a], j \in [b]\}$. Let $h : \mathcal{X} \setminus X \hookrightarrow \mathcal{Y} \setminus Y$ be a fixed 1–1 mapping, say the lexicographically smallest. The parametrized translator family is defined by,

$$\Theta' = \Theta_{m,n,Y} := \{-1,1\}^a, \quad f_\theta(x_{ij}) := \begin{cases} y_{ij}, & \theta_i = 1 \\ y_{i(j+1 \bmod b)}, & \theta_i = -1 \end{cases}, \quad \forall x \notin X \; f_\theta(x) = h(x).$$

Clearly $|\Theta'| = 2^a \le n = |\Theta|$ as needed. Let $\mathbf{x} = (x_1, x_2, \ldots, x_m)$. It suffices to show:

$$\Pr_{S,\mathbf{x}}\left[\mathrm{err}(\hat{\theta}) \ge \frac{\log n}{c_3 \alpha \min(10^5 m, |T|)}\right] \ge 0.99,$$

for some constant $c_3$ sufficiently large, because we can set $c_2 = 10^5 c_3$. The above equation is equivalent to the following two cases based on $m$:

$$\text{Case 1. } 10^5 m > |T| \implies \Pr_{S,\mathbf{x}}\left[\mathrm{err}(\hat{\theta}) \ge \frac{\log n}{c_3 \alpha |T|}\right] \ge 0.99 \tag{17}$$

$$\text{Case 2. } 10^5 m \le |T| \implies \Pr_{S,\mathbf{x}}\left[\mathrm{err}(\hat{\theta}) \ge \frac{\log n}{c_3 \alpha 10^5 m}\right] \ge 0.99 \tag{18}$$

In both cases, it will be convenient to notice that, for any $z \ge 2$, $\lfloor z \rfloor \ge z - 1 \ge \frac{z}{2}$. Since $b \ge 2$ (because $\alpha \le 1/2$), we therefore have

$$\frac{1}{2\alpha} \max\left(1, \frac{|T|}{10^5 m}\right) \le b \le \frac{1}{\alpha} \max\left(1, \frac{|T|}{10^5 m}\right) \tag{19}$$

**Case 1:** $10^5 m > T$. In this case $\frac{1}{2\alpha} \le b \le \frac{1}{\alpha}$ by Equation (19). We will show Equation (17). Now, consider the "full rows":

$$C(S) := \{i \in [a] \mid \forall j \in [b] \; y_{ij} \in S\}.$$

These rows will be useful to consider because nothing has been removed from the entire row, no information about $\theta_i$ has been revealed and (on average) one cannot achieve error $< 1/2$ on these examples, because one cannot distinguish between the two permutations on this row.

Note that the membership of different $i, i' \in C(S)$ is independent since $S$ is chosen independently, and by definition of $C$ and $S$:

$$\mathbb{E}[|C(S)|] = (1-\alpha)^b a \ge (1-\alpha)^{1/\alpha} a \ge \frac{a}{4},$$

since $(1-\alpha)^{1/\alpha}$ is decreasing in $\alpha$ and $\alpha \le 1/2$. Thus, by multiplicative Chernoff bounds (specifically, $\Pr[Z \le \mathbb{E}[Z]/2] \le e^{-\mathbb{E}[Z]/8}$),

$$\Pr_S\left[|C(S)| \le \frac{a}{8}\right] \le e^{-a/32} \le e^{-\lfloor c_1 \rfloor/32} \le 0.001, \tag{20}$$

for sufficiently large $c_1$. Thus, $\Pr_S[|C(S)| > a/8] \ge 0.999$. Let $C'(S, \mathbf{x}) \subseteq C(S)$ be those $i$ which $\hat{\theta}_i \ne \star_i$,

$$C'(S, \mathbf{x}) := \{i \in C(S) \mid \hat{\theta}_i \ne \star_i\}.$$

Clearly, for any algorithm and any $C(S)$, $\mathbb{E}_{\mathbf{x}}[|C'(S, \mathbf{x})| \mid S] = |C(S)|/2$ because no information whatsoever has been revealed about $\theta_i$ for any $i \in C$. Thus, by the same Chernoff bound, we have:

$$\Pr_{S,\mathbf{x}}\left[|C'(S, \mathbf{x})| \le \frac{1}{4}|C(S)| \;\middle|\; |C(S)| > \frac{a}{8}\right] \le e^{-\frac{a}{16} \cdot \frac{1}{8}} \le 0.001,$$

for sufficiently large $c_1$, because $a \ge c_1$. By the union bound over this and Equation (20),

$$\Pr_{S,\mathbf{x}}\left[|C'(S, \mathbf{x})| \ge \frac{a}{32}\right] \ge 0.998.$$

Since each row $i \in C'$ incurs $b$ errors on examples $x$, one for each $j$ because $f_\star(x_{ij}) \in S$:

$$\mathrm{err}(\hat{\theta}) \ge \frac{b \cdot |C'(S, \mathbf{x})|}{|T|}.$$

Thus,

$$\Pr_{S,\mathbf{x}}\left[\mathrm{err}(\hat{\theta}) \ge \frac{ba}{32\alpha|T|}\right] \ge 0.998.$$

Now, $a \ge \frac{1}{2}\log n$ for sufficiently large $c_1$ and as mentioned $b \ge \frac{1}{2\alpha}$. Thus,

$$\Pr_{S,\mathbf{x}}\left[\mathrm{err}(\hat{\theta}) \ge \frac{\log n}{128\alpha|T|}\right] \ge 0.998.$$

This establishes Equation (17) as long as $c_3 \ge 128$.

It remains to prove Equation (18).

**Case 2:** $10^5 m \leq T$. In this case $\frac{|T|}{2\alpha 10^5 m} \leq b \leq \frac{|T|}{\alpha 10^5 m}$ by Equation (19). Next, consider the set of rows with at least 1/2 of the elements in $S$:

$$D(S) := \left\{ i \in [a] \ \Big| \ |\{j \in [b] \mid y_{ij} \in S\}| \geq \frac{b}{2} \right\}.$$

Intuitively, any row $i \in D(S)$ is "dangerous" in the sense that if $\hat{\theta}_i \neq \star_i$, then it causes errors on $b/2$ different $x$'s in the support of $\mu$, i.e., for which $f_\star(x_{ij}) \in S$. Observe that $\mathbb{E}[|D(S)|] \geq a/2$ since each size $s = |\{j \in [b] \mid y_{ij} \in S\}| \geq b/2$ is at least as likely as the size $b - s$, since $\alpha \leq 1/2$. And also, membership of $i, i' \in D(S)$ since $S$ is independent. Thus, by the same Chernoff bound as above, for sufficiently large $c_1$,

$$\Pr_S\left[|D(S)| \leq \frac{a}{4}\right] \leq e^{-a/16} \leq e^{-\lfloor c_1 \rfloor/16} \leq 0.001. \tag{21}$$

Let $-\theta := (-\theta_1, -\theta_2, \ldots, -\theta_a)$. This makes it convenient to define the *giveaways* $G(S) \subseteq X$ to be,

$$G(S) := \{x_{ij} \mid i \in [a], j \in [b], f_\star(x_{ij}) \in S, f_{-\star}(x_{ij}) \notin S\}.$$

These are the points $x_{ij}$ which we might observe $f_\star(x_{ij})$ which would imply that $\theta_i = \star_i$ (and not its negative). Also let,

$$\hat{G}(S, \mathbf{x}) = \{x_1, x_2, \ldots, x_m\} \cap G(S).$$

(Note that if for a give row $i$, we do not have any $x_{ij} \in \hat{G}(S, \mathbf{x})$, then we have no information about $\theta_i$. As a preview to what is to come, we now argue that with high probability $|\hat{G}(S, \mathbf{x})| < a/8$ which will mean that, if $|D(S)| > a/4$, then we have no information about $\theta_i$ for at least $a/8$ of the rows $i \in D(S)$.)

For any fixed $i, j$, observe that $\Pr[x_{ij} \in G(S)] = \alpha(1 - \alpha)$ so $\mathbb{E}[|G(S)|] = \alpha(1 - \alpha)ab \leq \alpha ab$. By the Chernoff bound that $\Pr[Z \geq 2\mathbb{E}[Z]] \leq e^{-\mathbb{E}[Z]/3}$,

$$\Pr_S[|G(S)| \geq 2\alpha ab] \leq e^{-\alpha ab/3} \leq e^{-a/3} \leq 0.001,$$

for sufficiently large $c_1$. (We have used the fact that the above probability is smaller than if $E[|G(S)|]$ were actually $\alpha ab$.)

Also, $\mathbb{E}_S[|T \cap S|] \geq |T|/2$ since $\alpha \leq 1/2$. So, by the Chernoff bound $\Pr[Z \leq \mathbb{E}[Z]/2] \leq e^{-\mathbb{E}[Z]/8}$,

$$\Pr_S\left[|T \cap S| \leq \frac{|T|}{4}\right] \leq e^{-|T|/16} \leq 0.001,$$

for sufficiently large $c_1$ since $|T| \geq \log n \geq c_1$.

Thus, by the union bound:

$$\Pr_S\left[|T \cap S| \leq \frac{|T|}{4} \vee |G(S)| \geq 2\alpha ab\right] \leq 0.002. \tag{22}$$

Also,

$$\mathbb{E}_\mathbf{x}\left[|\hat{G}(S, \mathbf{x})| \mid S\right] \leq m\frac{|G(S)|}{|T \cap S|}.$$

Thus, using Markov's inequality in the second line below,

$$\mathbb{E}_{\mathbf{x},S}\left[|\hat{G}(S, \mathbf{x})| \ \Big| \ |T \cap S| > \frac{|T|}{4}, |G(S)| \leq 2\alpha ab\right] \leq m\frac{2\alpha ab}{|T|/4} = \frac{8\alpha abm}{|T|},$$

$$\Pr_{\mathbf{x},S}\left[|\hat{G}(S, \mathbf{x})| > \frac{8000\alpha abm}{|T|} \ \Big| \ |T \cap S| > \frac{|T|}{4}, |G(S)| \leq 2\alpha ab\right] \leq 0.001.$$

By the union bound over the above and Equation (22), since $\Pr[E] \leq \Pr[E|F] + \Pr[\neg F]$

$$\Pr_{\mathbf{x},S}\left[|\hat{G}(S, \mathbf{x})| > \frac{8000\alpha abm}{|T|}\right] \leq 0.001 + 0.002.$$

Finally, since

$$b \leq \frac{|T|}{\alpha 10^5 m}$$

$$\frac{8000 \alpha abm}{|T|} \leq 0.08 \cdot a \leq \frac{a}{8}$$

$$\Pr_{\mathbf{x},S}\left[|\hat{G}(S,\mathbf{x})| > \frac{a}{8}\right] \leq 0.003$$

By the union bound with Equation (21),

$$\Pr_{\mathbf{x},S}\left[|D(S)| \leq \frac{a}{4} \vee |\hat{G}(S,\mathbf{x})| > \frac{a}{8}\right] \leq 0.004$$

Let

$$F(S,\mathbf{x}) := \{i \in D(S) \mid \forall j \in [b] \; x_{ij} \notin \hat{G}(S)(S,\mathbf{x})\} \subseteq D.$$

Clearly $|F(S,\mathbf{x})| \geq |D(S)| - |\hat{G}(S,\mathbf{x})|$ because each $x \in \hat{G}(S,\mathbf{x})$ can remove at most one row from $D(S)$. Thus,

$$\Pr_{\mathbf{x},S}\left[|F(S,\mathbf{x})| \leq \frac{a}{8}\right] \leq 0.004. \tag{23}$$

$F(S,\mathbf{x})$ will function exactly like $C$ in the analysis above of Equation (17). We repeat this analysis for completeness, replacing $C$ by $F$. Let $F'(S,\mathbf{x}) \subseteq F(S,\mathbf{x})$ be those $i$ which $\hat{\theta}_i \neq \star_i$,

$$F'(S,\mathbf{x}) := \{i \in F(S,\mathbf{x}) \mid \hat{\theta}_i \neq \star_i\}.$$

For any algorithm and any $F(S,\mathbf{x})$, $\mathbb{E}_{\mathbf{x}}[|F'(S,\mathbf{x})| \mid S] = |F(S,\mathbf{x})|/2$ because no information whatsoever has been revealed about $\theta_i$ for any $i \in F$. Thus, by the same Chernoff bound, we have:

$$\Pr_{S,\mathbf{x}}\left[|F'(S,\mathbf{x})| \leq \frac{1}{4}|F(S,\mathbf{x})| \; \Big| \; |F(S,\mathbf{x})| > \frac{a}{8}\right] \leq e^{-\frac{a}{16}\cdot\frac{1}{8}} \leq 0.001,$$

for sufficiently large $c_1$. By the union bound over this and Equation (23),

$$\Pr_{S,\mathbf{x}}\left[|F'(S,\mathbf{x})| \geq \frac{a}{32}\right] \geq 0.995.$$

Since each row $i \in F'$ incurs $\geq b/2$ errors on examples $x$ by definition of $F'$ and $D$, since $F' \subseteq D$ and thus at least $b/2$ errors on $j$ for which $f_\star(x_{ij}) \in S$. Thus,

$$\mathrm{err}(\hat{\theta}) \geq \frac{b \cdot |F'(S,\mathbf{x})|}{2|T|}.$$

Thus,

$$\Pr_{S,\mathbf{x}}\left[\mathrm{err}(\hat{\theta}) \geq \frac{ba}{64\alpha|T|}\right] \geq 0.995.$$

Now, $a \geq \frac{1}{2}\log n$ for sufficiently large $c_1$ and we also have $b \geq \frac{|T|}{2\alpha 10^5 m}$ by Equation (19) since $\frac{|T|}{\alpha 10^5 m} \geq 2$ since $\alpha \leq 1/2$ and we are in the case where $10^5 m \leq |T|$. Thus,

$$\Pr_{S,\mathbf{x}}\left[\mathrm{err}(\hat{\theta}) \geq \frac{\log n}{256\alpha 10^5 m}\right] \geq 0.995.$$

This establishes Equation (18) for $c_3 \geq 256 \times 10^5$. $\qquad\square$

## D   Proofs for random knowledge graph

Our goal in this section is to prove Theorem 3.2. The proof is based on the following main lemma. We first state and prove this lemma, and then derive the theorem from it. Recall that the sets $T, P \subseteq \mathcal{Y} = Y \times Y = Y^2$ represent the edges of the two knowledge graphs.

**Lemma D.1.** *Fix $\emptyset \neq S \subseteq Y$, $\pi : Y^2 \hookrightarrow Y^2$, $\delta, p, \alpha \in (0,1)$, $q := 1 - p$, and*

$$\epsilon \geq 32 \cdot \frac{\ln(1/\delta)}{p\alpha^2 q^2 |S|^2}.$$

*For any $(T, P) \sim \mathrm{KG}(S, Y, p, \alpha)$ chosen from the random knowledge graph distribution, we define*

$$A := \{y \in T \mid \pi(y) \neq y\}$$
$$B := \{y \in A \mid \pi(y) \notin P\}$$
$$C := \{y \in A \mid y \notin P\}$$

*Then,*

$$\Pr_{T,P}\left[|A| \geq \epsilon|T| \ \wedge \ |B| - |C| \leq \frac{\alpha q}{2}|A|\right] \leq 5\delta.$$

*Proof.* Let $A' := \{y \in S^2 \mid \pi(y) \neq y\}$, so $A \subseteq A'$. If $\pi$ is the identity then the lemma holds trivially, therefore we can assume that $A' \neq \emptyset$. If $\epsilon > 1$ then the lemma holds trivially as well, because $A \subseteq T$ and therefore $|A| \leq |T|$, so we assume $\epsilon \in (0, 1]$.

For each $y \in A'$, Equations (1) and (2) under Definition 3.1 imply that $\Pr[y \in C] = \Pr[y \in T \setminus P] = (1 - \alpha)pq$ and $\Pr[y \in A] = \Pr[y \in T] = p$, thus Bayes rule gives

$$\forall y \in A' \ \ \Pr[y \in C \mid y \in A] = \frac{(1 - \alpha)pq}{p} = (1 - \alpha)q.$$

Now suppose we fix $V \subseteq A'$ and condition on $A := A' \cap T = V$. Note that the event $y \in C$ is independent for different $y \in V$, therefore for any $y \in V$ it holds that

$$\Pr_{T,P}[y \in C \mid A = V] = \Pr_{T,P}[y \in C \mid y \in A] = (1 - \alpha)q.$$

Therefore, $\mathbb{E}[|C| \mid A = V] = (1 - \alpha)q|A|$, and so a Chernoff bound gives

$$\forall V \subseteq A' \ \ \Pr_{T,P}\left[|C| \leq (1 - \alpha)q|A| + \sqrt{\frac{1}{2}|A|\ln\frac{1}{\delta}} \ \bigg| \ A = V\right] \geq 1 - \delta.$$

(Normally, Chernoff bounds would give the tight inequality that $\Pr[|C| < \ldots] \geq 1 - \delta$, but the $\leq$ in the above is necessary for the case in which $A = \emptyset$ in which case Chernoff bounds do not apply because it would be over $|A| = 0$ coin flips.) Since this holds for every $V$, we have:

$$\Pr_{T,P}\left[|C| \leq (1 - \alpha)q|A| + \sqrt{\frac{1}{2}|A|\ln\frac{1}{\delta}}\right] \geq 1 - \delta. \tag{24}$$

By Lemma C.2, we can partition $A'$ into three **disjoint** sets,

$$A' = A'_1 \cup A'_2 \cup A'_3 \text{ such that } \pi(A'_i) \cap A'_i = \emptyset.$$

As above, we are going to condition on the value of $A_i := A'_i \cap T$. Also, define,

$$B_i := \{y \in A_i \mid \pi(y) \notin P\} = B \cap A_i.$$

Now, fix any $i \in [3]$ and any set $V \subseteq A'_i$. We now claim that for all $i \in [3]$, $V \in A'_i$, and $y \in V$

$$\Pr_{T,P}[y \in B_i \mid A_i = V] = \Pr_{T,P}[\pi(y) \notin P \mid y \in T] = q.$$

The rightmost equality follows from the fact that $\pi(y) \neq y$ so $\pi(y) \notin P$ is independent of $y \in T$. The leftmost equality follows similarly: Since $\pi(A'_i) \cap A'_i = \emptyset$, the event $A_i = V$ is independent of $\pi(y) \notin P$. Thus, again by Chernoff bounds we have

$$\forall i \in [3] \ \forall V \subseteq A'_i \ \ \Pr_{T,P}\left[|B_i| \geq q|A_i| - \sqrt{\frac{1}{2}|A_i|\ln\frac{1}{\delta}} \ \bigg| \ A_i = V\right] \geq 1 - \delta.$$

Since this holds for all $V$, it holds unconditionally, and by the union bound it follows that

$$\Pr_{T,P}\left[\forall i \in [3] \ |B_i| \geq q|A_i| - \sqrt{\frac{1}{2}|A_i|\ln\frac{1}{\delta}}\right] \geq 1 - 3\delta. \tag{25}$$

Now, since the sets $B_i$ partition $B$ and $A_i$ partition $A$, we have $|B| = \sum_i |B_i|$, $|A| = \sum_i |A_i|$, and also $\sum_{i=1}^{3} \sqrt{|A_i|} \leq \sqrt{3|A|}$ by Cauchy–Schwartz. Thus, summing the three equations in Equation (25) probability implies

$$\Pr_{T,P}\left[|B| \geq q|A| - \sqrt{\frac{3}{2}|A|\ln\frac{1}{\delta}}\right] \geq 1 - 3\delta.$$

Combining with Equation (24) gives, by the union bound,

$$\Pr_{T,P}\left[|B| - |C| \geq q|A| - \sqrt{\frac{3}{2}|A|\ln\frac{1}{\delta}} - (1-\alpha)q|A| - \sqrt{\frac{1}{2}|A|\ln\frac{1}{\delta}}\right] \geq 1 - 4\delta.$$

Since $\sqrt{3/2} + \sqrt{1/2} \leq 2$, this implies:

$$\Pr_{T,P}\left[|B| - |C| \geq \alpha q|A| - 2\sqrt{|A|\ln\frac{1}{\delta}}\right] \geq 1 - 4\delta.$$

Or equivalently,

$$\Pr_{T,P}\left[|B| - |C| < \alpha q|A| - 2\sqrt{|A|\ln\frac{1}{\delta}}\right] \leq 4\delta.$$

Since adding additional restrictions can only reduce a probability, we have:

$$\Pr_{T,P}\left[\frac{p|S|^2}{2} \leq |T| \leq \frac{|A|}{\epsilon} \wedge |B| - |C| < \alpha q|A| - 2\sqrt{|A|\ln\frac{1}{\delta}}\right] \leq 4\delta.$$

But if $\frac{p|S|^2}{2} \leq |T| \leq \frac{|A|}{\epsilon}$ then $2|A| \geq \epsilon p|S|^2$ and then, since $\epsilon \geq \frac{32}{\alpha^2 q^2 p|S|^2}\ln\frac{1}{\delta}$:

$$2\sqrt{|A|\ln\frac{1}{\delta}} \leq 2\sqrt{\frac{2|A|}{\epsilon p|S|^2}\cdot|A|\ln\frac{1}{\delta}} \leq 2|A|\sqrt{\frac{2}{p|S|^2\frac{32}{\alpha^2 q^2 p|S|^2}\ln\frac{1}{\delta}}\ln\frac{1}{\delta}} = \frac{\alpha q}{2}|A|.$$

Thus,

$$\Pr_{T,P}\left[\frac{p|S|^2}{2} \leq |T| \leq \frac{|A|}{\epsilon} \wedge |B| - |C| < \frac{\alpha q}{2}|A|\right] \leq 4\delta.$$

Since, in general, for any two events $X$ and $Y$ it holds that $\Pr[Y] \leq \Pr[X,Y] + \Pr[\overline{X}]$, we have

$$\Pr_{T,P}\left[|T| \leq \frac{|A|}{\epsilon} \wedge |B| - |C| < \frac{\alpha q}{2}|A|\right] \leq \Pr_{T,P}\left[\frac{p|S|^2}{2} \leq |T| \leq \frac{|A|}{\epsilon} \wedge |B| - |C| < \frac{\alpha q}{2}|A|\right] +$$
$$+ \Pr_{T,P}\left[\frac{p|S|^2}{2} > |T|\right]$$
$$\leq 4\delta + \Pr_{T,P}\left[\frac{p|S|^2}{2} > |T|\right]$$
$$\leq 4\delta + \delta,$$

which is equivalent to the statement in the lemma. To see the last step above, note that $\mathbb{E}[|T|] = p|S|^2$ and thus by multiplicative Chernoff bounds,

$$\Pr\left[|T| < \frac{p|S|^2}{2}\right] \leq \exp\left(-\frac{p|S|^2}{8}\right) \leq \exp\left(-\frac{p|S|^2}{8}\cdot\frac{1}{\epsilon}\cdot\frac{32}{\alpha^2 q^2 p|S|^2}\ln\frac{1}{\delta}\right) = \delta^{\frac{4}{\epsilon\alpha^2 q^2}} \leq \delta.$$

In the last step we have utilized the fact that $\alpha, q, \delta \in (0,1]$, and the fact (observed in the first paragraph of this proof) that we may assume that $\epsilon \in (0,1]$ else the lemma holds trivially. $\square$

Using the above lemma, we now prove our main theorem regarding knowledge graphs.

*Proof.* of Theorem 3.2. Let $q := 1 - p$ and,

$$\epsilon := \max\left(\frac{64}{\alpha^2 p q^2 |S|^2} \ln \frac{6n^{|S|}}{\delta}, \frac{2}{\alpha q} \sqrt{\frac{2}{m} \ln \frac{6n^{|S|}}{\delta}}\right).$$

For any $\theta \in \Theta$ define,

$$A_\theta := \{y \in T \mid \pi_\theta(y) \neq y\}$$
$$B_\theta := \{y \in A_\theta \mid \pi_\theta(y) \notin P\}$$
$$C_\theta := \{y \in A_\theta \mid y \notin P\}$$

Note that since $\text{err}(\theta) = |A_\theta|/|T|$, our goal is to show that, with probability $\geq 1 - \delta$, we will not output any $\theta$ with $|A_\theta| \geq \epsilon|T|$.

Recall that $|\Theta| \leq n^{|S|}$. By Lemma D.1 substituting $\delta' = \frac{1}{6n^{|S|}}\delta$ and the union bound over $\theta \in \Theta$ which is of size $|\Theta| \leq n^{|S|}$,

$$\Pr_{T,P}\left[\exists \theta \in \Theta \ |A_\theta| \geq \epsilon|T| \wedge |B_\theta| - |C_\theta| \leq \frac{\alpha q}{2}|A_\theta|\right] \leq \frac{5\delta}{6}.$$

Using $\text{err}(\theta) = |A_\theta|/|T|$, this implies,

$$\Pr_{T,P}\left[\exists \theta \in \Theta \ \text{err}(\theta) \geq \epsilon \wedge |B_\theta| - |C_\theta| \leq \frac{\alpha q \epsilon |T|}{2}\right] \leq \frac{5\delta}{6}$$
$$\Pr_{T,P}\left[\exists \theta \in \Theta \ \text{err}(\theta) \geq \epsilon \wedge \frac{|B_\theta|}{|T|} - \frac{|C_\theta|}{|T|} \leq \frac{\alpha q \epsilon}{2}\right] \leq \frac{5\delta}{6} \qquad (26)$$

Finally, define the empirical "errors" for any $\theta$ to be,

$$\hat{e}_\theta = \frac{1}{m}\{i \mid f_\theta(x_i) \notin P\}.$$

It is not difficult to see that the algorithm outputs a $\theta$ with minimal $\hat{e}_\theta$, and thus it will not output any $\theta$ with $\hat{e}_\theta - \hat{e}_\star > 0$. Now, it is also not difficult to see that $\hat{e}_\theta - \hat{e}_\star$ is the mean of $m$ random variables in $\{-1, 0, 1\}$ and

$$\mathbb{E}[\hat{e}_\theta - \hat{e}_\star] = \Pr_{y \sim \tau}[\pi_\theta(y) \notin P] - \Pr_{y \sim \tau}[y \notin P] = \frac{|B_\theta|}{|T|} - \frac{|C_\theta|}{|T|}.$$

The last step above follows because $\pi_\star$ is the identity, and because if $y = \pi_\theta(y)$ then $y \in P \iff \pi_\theta(y) \in P$. (Formally, one may define $E_\theta := \{y \in T \mid \pi_\theta(y) \notin P\}$ and observe that $B_\theta \subseteq E_\theta$, $C_\theta \subseteq E_\star$ and $E_\theta \setminus B_\theta = E_\star \setminus C_\theta$). Thus, by Chernoff bounds,

$$\forall \theta \in \Theta \quad \Pr_{x_1,\ldots,x_m}\left[\hat{e}_\theta - \hat{e}_\star \leq \frac{|B_\theta|}{|T|} - \frac{|C_\theta|}{|T|} + \sqrt{\frac{2}{m} \ln \frac{6|\Theta|}{\delta}}\right] \leq \frac{\delta}{6|\Theta|}.$$

By the union bound over $\theta \in \Theta$,

$$\Pr_{x_1,\ldots,x_m}\left[\exists \theta \in \Theta \ \hat{e}_\theta - \hat{e}_\star \leq \frac{|B_\theta|}{|T|} - \frac{|C_\theta|}{|T|} + \sqrt{\frac{2}{m} \ln \frac{6|\Theta|}{\delta}}\right] \leq \frac{\delta}{6}.$$

Combining with Equation (26) gives,

$$\Pr_{T,P}\left[\exists \theta \in \Theta \ \text{err}(\theta) \geq \epsilon \wedge \hat{e}_\theta - \hat{e}_\star \leq \frac{\alpha q \epsilon}{2} - \sqrt{\frac{2}{m} \ln \frac{6|\Theta|}{\delta}}\right] \leq \frac{5\delta}{6} + \frac{\delta}{6} = \delta.$$

Since, for our choice of $\epsilon \geq \frac{2}{\alpha q}\sqrt{\frac{2}{m} \ln \frac{6|\Theta|}{\delta}}$,

$$\Pr_{T,P}[\exists \theta \in \Theta \ \text{err}(\theta) \geq \epsilon \wedge \hat{e}_\theta - \hat{e}_\star \leq 0] \leq \delta.$$

Put another way,

$$\Pr_{T,P}\left[\forall \theta \in \Theta \ \ \mathrm{err}(\theta) \le \epsilon \vee \hat{e}_\theta - \hat{e}_\star > 0\right] \ge 1 - \delta.$$

We claim that we are done: Observe that if MLE outputs some $\theta \ne \star$ then, $\hat{b}_\theta \le \hat{b}_\star$. To see this, recall the definition of the prior $\rho$,

$$\rho(f_\theta(x)) := \begin{cases} \frac{1}{2} \cdot \left(\frac{1}{|P|} + \frac{1}{|\mathcal{Y}|}\right) & \text{if } f_\theta(x) \in P \\ \frac{1}{2|\mathcal{Y}|} & \text{if } f_\theta(x) \notin P. \end{cases}$$

and therefore the objective function minimized by MLE, namely, $\frac{1}{m}\sum_{i=1}^m -\log(\rho(f_\theta(x_i)))$, is strictly monotonic in $\hat{b}_\theta$:

$$\frac{1}{m}\sum_{i=1}^m -\log(\rho(f_\theta(x_i))) = \hat{b}_\theta \cdot \log \frac{2}{1/|\mathcal{Y}|} + (1 - \hat{b}_\theta)\log \frac{2}{1/|P| + 1/|\mathcal{Y}|}$$

$$= \log \frac{2}{1/|P| + 1/|\mathcal{Y}|} + \hat{b}_\theta \cdot \log \frac{1/|P| + 1/|\mathcal{Y}|}{1/|\mathcal{Y}|}$$

so the $\theta$ output by MLE necessarily minimizes $b_\theta$.

Finally, for the simplification in the theorem, note that for $p < 0.99$, $1/q < 100$ is at most a constant and note that a maximum is never more than a sum. $\qquad\square$

It is interesting to note that it is possible to prove the same theorem using a generalization of Plausible Ambiguities, though we use the shorter proof above here because it is somewhat more involved. This generalization may be useful for other priors of full support. Many LMs, in practice, assign non-zero probability to every string due to softmax distributions or a process called "smoothing." A full-support prior $\rho$ has full support, then $\mathcal{A}_\gamma = \Theta$ and so the parameter $\varepsilon_\gamma$ becomes too large to be meaningful even for $\gamma = 0$. To address this, we refine our definition of plausible ambiguities as follows. For generality, we state them in terms of arbitrary loss $\mathcal{L}$, though we only use them for the semantic error $\mathcal{L} = \mathrm{err}$.

**Definition D.2** $((\gamma, \kappa)$-plausible ambiguities)**.** *For any $\gamma, \kappa \in [0,1]$, the set of $(\gamma, \kappa)$-plausible ambiguities is:*

$$\mathcal{A}_{\gamma,\kappa} := \left\{ \theta \in \Theta \ \middle| \ \Pr_{y \sim \tau}\left[\rho(\pi_\theta^\star(y)) \le \kappa\right] \le \gamma \right\}, \ and \ \varepsilon_{\gamma,\kappa} := \max_{\theta \in \mathcal{A}_\gamma} \mathcal{L}(\theta).$$

*Furthermore, $\mathcal{A}_\gamma = \mathcal{A}_{\gamma,0}$ and $\varepsilon_\gamma = \varepsilon_{\gamma,0}$.*

# E   Experiments

For the experiments used to generate Figures 4 and 5, we sampled random languages according to either the knowledge graph model or the common nonsense model, and then used a brute-force learning algorithm to find the optimal translator given an increasing amount of samples. A detailed description follows, and code can be found at `https://github.com/orrp/theory-of-umt`.

## E.1   Experiments in the knowledge graph model

Recall that in the knowledge graph model, text describes relations between nodes in a directed graph. Due to computational constraints, we consider ten nodes, each corresponding to a different word in the target language. To generate edges corresponding to the target language $P$, two nodes are connected with a directed edge independently, with probability $0.5$. We then consider source languages with $r \le 10$ words. Given a ground-truth translator $f_\star \colon [r] \to [10]$, the source language graph $T$ is obtained by choosing a random subset of nodes $S$ of size $r$, taking the pre-image of graph induced on $S$ under $f_\star$, and (3) adding noise by redrawing each edge with probability $1 - \alpha$ for a fixed agreement coefficient $\alpha \in (0,1)$.

The prior $\rho$ is derived from the edges of $P$, and the source language $\mu$ is derived from the (noisy) permuted subgraph $T$. We consider the translator family $\{f_\theta | \theta \in \Theta\}$ of all node-to-node (word-to-word) injective translators, of which one is secretly chosen to be ground-truth. Similarly to the

| Name | Symbol | Value |
|------|--------|-------|
| Number of source nodes | $r$ | $1, 4, 7, 10$ |
| Number of target nodes | $n$ | $10$ |
| Number of training data | $m$ | $1, 2, \ldots$ up to all edges |
| Edge density (probability of including an edge) | $p$ | $0.5$ |
| Agreement parameter | $\alpha$ | $0, 0.33, 0.66, 1$ |

Figure 7: Parameters for experiments in the knowledge graph model (Figure 4). For ablations on $r$ we take $\alpha = 0.5$, and for ablations on $\alpha$ we take $r = 9$. The experiments were run in parallel on an AWS r6i.4xlarge for a total of two and a half CPU-hours.

| Name | Symbol | Value |
|------|--------|-------|
| Number of source sentences | $|T|$ | $10^5$ |
| Number of target sentences | $|P|$ | $10^6$ |
| Number of training data | $m$ | $1, \ldots, 100$ |
| Number of validation data | | $1000$ |
| Fraction of common nonsense | $\alpha$ | $0, 0.1, \ldots, 0.8$ |

Figure 8: Parameters for experiments in the common nonsense model (Figure 5). The experiments were run in parallel on an AWS r6i.4xlarge for a total of four CPU-hours.

previous setting, we train an MLE algorithm on randomly chosen edges from $T$, which correspond to sentences in the source language. For each sampled edge $(x_1, x_2)$, we increase the "score" of each translator that agrees with the edge, that is, that $(f_\theta(x_1), f_\theta(x_2))$ is an edge in the graph $P$.

To show how common ground affects translatability, we ablate the parameter $\alpha$ determines the fraction of edges on which the source language graph $T$ and the target language graph $P$ agree. Figure 4 validates the intuition that increased agreement results in lower translation error, and that as the number of samples increases, the error of the top-scoring translator decreases.

To show how language complexity affects translatability, we ablate $r$, which is the size of the subgraph corresponding to the source language. Figure 4 (right) validates the intuition that a larger subgraph results in lower translation error.

The error of a translator is computed as the fraction of edges whose labels are different than the ground-truth. The values with which the model is instantiated are detailed in Figure 7.

### E.2 Experiments in the common nonsense model

Since in this model the structure of sentences is arbitrary, we represent sentences by integer IDs, $[10^5] = 1, 2, \ldots, 10^5$ and $[10^6]$ for the target language. We generate a prior $\rho$ from the common nonsense model by taking the target sentence ids $[10^6]$ and labeling a random $\alpha$-fraction of them as nonsense; the remaining sentences are called sensical $S$. Given a ground-truth translator $f_\star : [10^5] \to [10^6]$, the source language then distributes uniformly over the back-translation of sensical sentences, $f_\star^{-1}(S)$.

The translator family $\{f_\theta | \theta \in \Theta\}$ is taken to be a set of $10^5$ random one-to-one translators, of which one is secretly chosen to be ground-truth $f_\star$. We then train an MLE algorithm on random samples from the source language: Each sample $x \sim \mu$ rules-out a subset of translators, namely, all $\theta \in \Theta$ such that $f_\theta(x) \notin S$, i.e., is nonsensical.

Figure 5 shows that as the number of samples increases, the average error over the plausible translators (that have not been ruled-out) decreases. To show how language complexity / common ground affect translatability, we ablate the parameter $\alpha$ which determines the fraction of common nonsense. Our experiments validate the intuition that increased common nonsense results in lower translation error. The error of a translator is computed as the fraction of disagreements with the ground-truth on a hold-out validation set of size 1000. The values with which the model is instantiated are detailed in Figure 8.

# F   Related work

**Project CETI.**   The sperm whale data collection effort began with a longitudinal dataset from a community of whales off the coast of Dominica that revealed interesting communication findings, such as dialects and vocal clans [16]. A recent effort by the Cetacean Translation Initiative (Project CETI) has been to collect custom-built passive bioacoustic arrays (installed in Fall 2022) covering a $20 \times 20$ kilometer area where these whale families reside (collecting over 3 TB/month) in tandem with on-whale robotic acoustic and video tags, underwater (robotic swimming fish) and aerial drones as well as other observation techniques in effort to augment rich contextual communication data. CETI's scientific team consists of specialists in machine learning, robotics, natural language processing, marine biology, linguistics, cryptography, signal processing and bio-acoustics. Andreas et al. [2] present CETI's initial scientific roadmap for understanding sperm whale communication, identifying the potential for unsupervised translation to be applied to whale communication. That roadmap suggests training a full generative LM for whale communication (often using trillions of bits for parameters [11, 12]). In contrast, our analysis suggests that the data requirements for translation may be similar to those of supervised translation, which is often several orders of magnitude smaller [37].

With this setting in mind, our requirements from source and target language are not symmetric: it would be unreasonable (and unnecessary) for our framework to assume that any sentence in the target language could be expressed in the source language. Put simply: whales need not understand what a smartphone is for us to gain some understanding of their communication. Also note, regarding domain gap, that some (although not all) knowledge can be inferred by training data from, e.g., online catalogs of hundreds of thousands of marine species [1]). Of course, there are also data-collection and transcription challenges, a challenge also present in the setting of *low-resource* (human) language translation [31]. While these challenges are outside the scope of this paper, our theoretical bounds on the data requirements may inform how much and what types of data are collected. For instance, it is less expensive to acquire textual data alone than both textual and video data. Therefore, if it is believed that an adequate translation is statistically possible using textual data alone, then greater effort may be placed on collecting this data and on UMT algorithms.

**Unsupervised translation.**   In unsupervised machine translation [32], a translator between two languages is learned based only on monolingual corpora from each language. A body of work on UMT uses neural networks [29, 23, 5, 24, 35] or statistical methods [25, 4] for this task. Empirical evaluation of UMT found that it is outperformed by supervised machine translation, even when UMT is trained on several orders of magnitude more data [28, 22]. Among the key barriers for UMT identified in these evaluations are the domain gap and the data gap, and recent works propose techniques for bridging these gaps [15, 19]. Our theory suggests that sample complexity should remain roughly the same between the supervised and unsupervised settings, barring computational constraints. This, we hope, will encourage practitioners to bridge the remaining gaps.

**Language models (LMs).**   In recent years, LMs such as GPT [11], BERT [13] and PaLM [12] were shown to achieve state-of-the-art performance on many tasks in natural language processing (NLP) such as text generation, summarization, or (supervised) MT. These models are indeed large, with hundreds of billions of parameters, and are pre-trained on hundreds of billions of tokens.

LMs are useful for machine translation in a variety of ways (e.g. [10, 18]). Of particular relevance are empirical works that use target LMs as priors to improve machine translation [27, 7]. To our knowledge, our work is the first *theoretical* work formally proving error bounds for prior-assisted translation. Appendix H discusses the use of LMs to establish priors for translation.

**Goal-oriented communication.**   It is interesting to contrast our work with the work on goal-oriented communication, which was introduced by [21] and extended in [17]. They study the setting of two communicating parties (one of which is trying to achieve a verifiable goal) using each a language completely unknown to the other. They put forward a theory of goal-oriented communication, where communication is not an end in itself, but rather a means to achieving some goals of the communicating parties. Focusing on goals provides a way to address "misunderstanding" during communication, as in when one can verify whether the goal is (or is not) achieved. Their theory shows how to overcome any initial misunderstanding between parties towards achieving a given goal. Our setting is different: Informally, rather than be a participant in a communication with someone

| Sentence | Probability |
|---|---|
| I just ate a giant squid. | $1.5 \times 10^{-9}$ |
| I just ate a giant cheeseburger. | $\mathbf{1.9 \times 10^{-8}}$ |
| A sperm whale said: I just ate a giant squid. | $\mathbf{6.8 \times 10^{-14}}$ |
| A sperm whale said: I just ate a giant cheeseburger. | $1.2 \times 10^{-17}$ |

Figure 9: Without using a prompt, the sentence *I just ate a giant cheeseburger* is more likely, but using the prompt *A sperm whale said:*, the sentence *I just ate a giant squid* is much more likely. Probabilities are from the GPT-3 API.

speaking a different language, we wish to translate communications between two external parties speaking in a language unknown to us and there is no verifiable goal to aid us in this process.

**Subgraph isomoprhism.** For simplicity, we model the knowledge graphs of Section 3.1 as a pair of correlated Erdős–Rényi (ER) graphs. The computational problem of identifying a subgraph of an ER graph has been studied by Erdős and others [6, 26, 20]. In particular, [20] consider a model in which two correlated graphs $P, T$ are derived from a "parent graph" $G$ by independently deleting rows and edges $G$, and then applying a permutation $\pi^*$ to the vertices of $T$. Although their model differs from our knowledge graph model,[15] they propose efficient algorithms for recovering the latent permutation $\pi^*$ and provide an empirical evaluation on synthetic and real-world data. Given the similarity between our models, it would be interesting to see if their algorithm can be adapted to our setting, which would nicely complement our formally-proven-yet-inefficient algorithm.

# G   Future Work

Our initial exploration leaves plenty of room for future work. In particular, we propose the following possible directions:

1. In our lossless models, the target language subsumes the source language in the sense that everything that is representable in the source language can also be represented in the target language. It would be interesting to extend our work to the partially-overlapping case.

2. The language distribution in our models are all uniform. It would be interesting to examine non-uniform distributions such as Zipfian or power-law distributions.

3. As stated earlier, our analysis is purely information-theoretic and leaves the question of the *efficiency* of UMT open.

4. A good starting point for the efficiency question would be to design efficient UMT algorithms for one of the randomized models of language presented in this paper.

# H   Good Priors through Prompts

The most direct way to improve a prior is to train (or fine-tune) the LM on a dataset that includes a large number of articles relevant to the source language, e.g., volumes of oceanic research, for whales. In addition, training on a wide variety of sources including multiple languages, diverse sources, and encoding systems may be helpful. It is possible that a system that has how to transfer knowledge between hundreds of languages and even programming languages, may have a better prior.

Another general strategy for creating a prior is to use prompting: Given a *prompt* string $s$, one can define $\rho(y) \propto \nu(s\,y)$, that is the prior distribution of text that that LM generates conditioned on the text beginning with $s$. Figure 9 illustrates some toy examples of how even a simple prompt like *A sperm whale said:* can help focus on translations that are more likely for a sperm whale to say, and eliminate irrelevant translations.

---

[15]In the knowledge graph (a) the vertices of $T$ are *always* a subset of the vertices of $P$, and (b) the deleted vertices are fixed rather than randomly chosen

**Background prompts.** There is a natural and potentially powerful idea that an unsupervised translator, in addition to outputting translations, would automatically generate a *background prompt* that increases the intelligibility of many translations.[16] Suppose, for example, across numerous communications, the unsupervised translator determines that sperm whales measure time in "number of naps" and that typical nap duration varies by age. Then, rather than having to repeatedly explain this in each translation, it can be explained once in a background prompt that is automatically inserted before each translation. For example, following the prompt

$s =$ *Sperm whales measure time in numbers of naps, but each whale's typical nap duration depends on their age. So if a whale's nap duration is 9 minutes, then 4 naps is about 36 minutes (though whales tend to exaggerate times). A sperm whale said:*

would make translations like *Wow, I just dove for 6 naps* or *How many naps ago was that?* both more likely and more understandable.

# I  Generalizations of the framework

## I.1  Lossy Translation

Many things can be included in textual transcriptions suitable for translation. For instance, one can distinguish the speakers, e.g., "*Whale 1: ... Whale 2: ... Whale 1: ...*" if the source data is annotated with speaker identifiers. Some aspects of the way one is speaking can be transcribed, e.g., "*Whale 1 (fast tempo clicking): ... Whale 2 (slow, loud clicking): ...*" It may be possible to encode these textually in $x$.

However, if $x$ is encoded in a more flexible format, such as arbitrary binary files, one can include as much raw information as possible, including the source audio recording, to provide the translator with as much context as possible. In that case, lossless translation will no longer be possible, because one cannot compute the raw $x$ from a textual translation.

Given the possible benefits of such annotations, we propose an extension of our theory to the lossy setting. A natural generalization of the maximum-likelihood approach is as follows:

$$\min_{\theta \in \Theta} \frac{1}{m} \sum_{i=1}^{m} -\log \rho(f_\theta(x_i)) - \frac{1}{\lambda} \log \phi_\theta(x_i \mid y = f_\theta(x_i)).$$

Here $\phi_\theta \colon \mathcal{X} \times \mathcal{Y} \to [0, 1]$ is a probabilistic inverse ("randomized back translation") of $f$ whose parameters are also encoded in $\theta$. Note that the family $\{(f_\theta, \phi_\theta) \mid \theta \in \Theta\}$ must satisfy that for all $y \in \mathcal{Y}, \sum_{x \in f^{-1}(y)} \phi_\theta(x \mid y) = 1$, though it is no longer required that $f_\theta$ be 1–1.

As $\lambda$ decreases to 0, the optimal solution would assign infinite loss to any $f_\theta \colon \mathcal{X} \hookrightarrow \mathcal{Y}$ and $\phi_\theta$ that are not perfect inverses, where there is some $x$ (with positive probability under $\mu$) such that $\phi_\theta(x \mid y = f_\theta(x)) < 1$. Thus, for sufficiently small $\lambda$, the algorithm is exactly the minimum cross-entropy (maximum likelihood) algorithm of Definition 1.1. For arbitrarily large $\lambda$, the algorithm will collapse to always outputting the most likely $y \in \mathcal{Y}$ under $\rho$. For example, everything could be translated to *Hello* regardless of its contents.

For intermediate values of $\lambda$, the chosen translator trades off naturalness in the form of $\rho(f_\theta(x))$ versus information loss which is inversely related to $\phi_\theta(x_i \mid y = f_\theta(x_i))$. This trade-off makes it challenging to define and analyze the success of a lossy unsupervised translation algorithm. Nonetheless, the algorithm is intuitive.

## I.2  Infinite parameter sets $\Theta$

In some cases, one can learn with many fewer examples, which is important when parameters are real-valued and $|\Theta| = \infty$ or when the model is over-parameterized. However, one can analyze even these cases in our model using the following "trick." Suppose one has a supervised translation algorithm SUPER that takes $m$ labeled examples $(x_i, y_i)$ as input and outputs $\theta \in \Theta$. A simple

---

[16]In general, the problem of AI-based prompt generation has recently attracted attention, e.g., [34].

observation in these cases is that one could use an initial set of $m$ unlabeled examples, $x_1, \ldots, x_m$, to define a subset of translators:

$$\overline{\Theta} := \left\{ \text{SUPER}\big((x_1, y_1), (x_2, y_2), \ldots, (x_m, y_m)\big) \mid y_1, y_2, \ldots, y_m \in \mathcal{Y} \right\}.$$

That is, we restrict attention to the set of possible translators that we could output for any given ground-truth translations, then it is not difficult to see $\log |\overline{\Theta}| \leq m \log |\mathcal{Y}|$. If we assume that one of these is accurate and natural, then restricting the attention of MLE to this set will suffice, and Theorem A.2 means that the number of examples required is $O(\log |\overline{\Theta}|) = O(m \log |\mathcal{Y}|)$ which is a linear blowup in the number of examples $m$ used for supervised translation. To make this formal, one would start from only assuming that one of the translators had negligible error—this is left to future work.

## J  Bounds for Supervised Translation

In this work, for ease of presentation, we have stated that error decreases like $O(\frac{1}{m} \log \frac{|\Theta|}{\delta})$ for noiseless supervised translation. This is based in the classic Occam bound for supervised learning:

**Theorem J.1** (Occam bounds). *Let $\mathcal{X}, \mathcal{Y}$, be sets, $\mathcal{D}$ be a joint distribution over $\mathcal{X} \times \mathcal{Y}$, $\ell \colon \mathcal{Y} \times \mathcal{Y} \to [0, 1]$ be a loss, and $f_\theta : \mathcal{X} \to \mathcal{Y}$ be a family of functions parameterized by $\theta \in \Theta$, and $\mathcal{L}(\theta) := \mathbb{E}_{(x,y) \sim \mathcal{D}}[\ell(y, f_\theta(x))]$. For any $\delta > 0$,*

$$\Pr_{(x_1, y_1), \ldots, (x_m, y_m) \sim \mathcal{D}^m} \left[ \mathcal{L}(\hat{\theta}) \leq \frac{1}{m} \ln \frac{|\Theta|}{\delta} \right] \geq 1 - \delta \qquad\qquad \text{if } \mathcal{L}(\star) = 0,$$

$$\Pr_{(x_1, y_1), \ldots, (x_m, y_m) \sim \mathcal{D}^m} \left[ \mathcal{L}(\hat{\theta}) \leq \mathcal{L}(\star) + \sqrt{\frac{1}{m} \ln \frac{|\Theta|}{\delta}} \right] \geq 1 - \delta \qquad \text{if } \mathcal{L}(\star) \neq 0,$$

*for $\hat{\theta} := \operatorname{argmin}_{\theta \in \Theta} \sum_i \ell(y_i, f_\theta(x_i))$.*[17]

Note that supervised translation is simply classification with many classes $\mathcal{Y}$.

---

[17]As in MLE, ties can be broken arbitrarily, e.g., lexicographically. Our bounds, like the Occam bound, hold simultaneously for all minimizers. In the realizable case, the Empirical Risk Minimizer $\hat{\theta}$ will have $\sum \ell(y_i, f_\theta(x_i)) = 0$.

