# OpenReview forum: "A Theory of Unsupervised Translation Motivated by Understanding Animal Communication"
_NeurIPS.cc/2023/Conference — NeurIPS 2023 poster_

### Official Review · Reviewer_FHLH · 2023-06-11

**Soundness:** 2 fair
**Presentation:** 1 poor
**Contribution:** 3 good
**Rating:** 3
**Confidence:** 3

**Summary:**

This paper proposes a theoretical framework for understanding unsupervised translation and provides bounds for sample complexity. Specifically, the unsupervised machine translation setting they consider is that the translator $f_\theta$ is the one that maximizes the plausibility $\rho$ of the translation of examples $x_1, x_2, \cdots, x_m$ in the source language:

$$\theta = \mathrm{MLE}^{\rho}(x_1, x_2, \cdots, x_m) = \arg \max_{\theta} \sum_{i=1}^m \log \rho(f_\theta(x_i))$$

The framework they propose has the following key components:

- Prior $\mu$: the distribution of the source language.
- Prior $\rho$: the plausibility of the translation in the target language.
- Translator: $f_\theta$ which they assume belongs to a finite-sized family.
- Translation density $\tau$: The distribution induced by the translator, i.e. $f_\theta{x}$ where $x \sim \mu$.
- A ground truth translator: $f_*$ that maps any input $x$ to a plausible translation almost surely.
- Semantic loss: $L(\theta)$ that evaluates how the translations by $f_\theta$ is correct. They use exactly match (0-1 loss) as $L$ in the following analysis.

As instantiations of their framework, they propose two models of language:

## Random knowledge graph model

This model makes the following assumptions:
1. Text in this language represents an edge between a pair of nodes in a knowledge graph.
2. The translator translates an edge by translating the two nodes respectively.
3. The plausible edges in the source language largely agree with the plausible edges in the target language.
Based on these assumptions the authors construct the distributions $\rho, \tau \mu$, and then they provide the sample complexity bound for this model.


## Common nonsense model

This model assumes no linguistic structure on the source text. They only assume that there is a set of translations that are implausible. They provide a sample complexity bound for this model. They also make connections to the complexity of the translator (description length).


**Strengths:**

1. In my opinion, this paper’s assumption about the existence of a set of plausible translations is very realistic and insightful for understanding the mechanism of unsupervised translation. As far as I know, attributing the success of unsupervised translation to this characteristic is novel and I think it is convincing (though may not be a complete explanation).
2. This paper also demonstrates how this assumption can be used to analyze unsupervised translation.
3. I also like the consideration about having a plausible prior in addition to the ground truth translation distribution. It handles the mismatch between the domains of the text in the source and target language. I think this kind of domain mismatch is very common in practice and thus considering this helps us understand unsupervised translation better.
4. The knowledge graph model is inspiring to me. I think it suggests that the success of unsupervised translation is because of the interaction between some high-level structure and some low-level structure of language. In this model, the high-level structure is the plausibility of the combination of nodes, while the low-level structure is the one-to-one mapping of the nodes (if I understood correctly).

In sum, I think I learned quite a few things from this paper. I think the community can benefit from these ideas.


**Weaknesses:**

## Clarity Issues

The first major issue to me is that this paper has a very unusual structure, i.e. having 5 pages of introduction.
1. I would expect the introduction to talk more about the motivation and high-level intuition of the theory. However, the authors tried to elaborate many definitions in the introduction but at the same time mention a few things without clear definition. For example, I couldn’t understand “broader prior (line 69)” and “agreement (line 99)” when I was reading the introduction.
2. Another problem of this long introduction is that there seems to be a lot of overlap between the introduction and the content in the following sections. Information in the later sections does not seem to be totally aligned with the description in the introduction section. For example, at line 101, the authors assume the compositional encoding of language, but this is not mentioned in Section 3.1.

The second  issue is that this paper never provides a high-level explanation for the proofs in the main text. It’s hard to assess the correctness of their results. I found the proofs in the appendix are not easy to follow either.

The third issues is that they rarely provide explanations for the symbols or the formulas either. For example, in Definition 3.1, they define many sets (e.g. $P, S$ etc) without describing their intuitive meanings. The usage of the symbols is also not very intuitive. For example, it seems that the $T, S$ and $P$ in Section 3.1 have different meanings than the ones in Section 3.2.

## Technical Issues

I have two main concerns:

1. I doubt whether Theorem 3.4 is correct. Intuitively speaking, based on this model, there are many translators that never map any source text to an implausible target text. However, only one of them is correct. How can it be possible that the translator learned from MLE can be a good one without the access to other information? I also checked the proof in Appendix C.1.1. Line 714 seems to be saying it is suffice to prove that “large semantic error ($A$) implies many implausible translations ($B$)”, but the formula in Line 717 is more like saying “it’s impossible to have low error ($\neg A$) while having many implausible translations ($B$)”. If the goal is to prove $A \to B$, then I can’t see why we should prove $( \neg A \wedge B)$ is impossible.
2. I think there needs to be more justification for how the instantiations are related to real world scenarios. One apparent example is the distributions of $\rho$ and $\nu$ constructed based on Definition 3.1, which is very complicated and is not intuitive. I would like to know whether this is also an abstraction of real-world scenarios.

In sum, I think this work is not ready for publication in this form, despite the interesting insights it has.


**Questions:**

1. Please let me know if I misunderstood anything about the technical issues. I would be happy to adjust my score.
2. Could you describe the meaning of the broader prior $\rho$ and agreement $a$ in the context of real-world scenarios? e.g. what would we expect this $\rho$ and $a$ to be in the real world?
2. I wonder whether the back translation loss used in some unsupervised translation works is relevant to the one-to-one translator assumption in this work? Could it be helpful because the loss enforces the bijectivity of the model?
3. I would also suggest the authors to include some experiments, perhaps with some toy data.


**Limitations:**

In my opinion, the authors should list their assumptions more explicitly and clearly, e.g. including them in a list or indexing them with numbers. Other than that, I think the discussion section addresses the limitations well.

---

> ### Author Rebuttal · Authors · 2023-08-10
>
> > **I doubt whether Theorem 3.4 is correct.** ...
>
> You indeed found a typo in our proof overview intuition in section C.1.1. However, the theorem and proof are correct. We apologize for the mistake on line 717 and will correct it. Thank you for identifying it! Let us explain first the intuition and then the typo.
>
> Your intuition is correct when $\Theta$ includes all mappings from $\mathcal{X} \hookrightarrow \mathcal{Y}$, i.e., there is no structural assumption to the translation whatsoever. In this case, our bounds are vacuous. However, if the ideal translator is assumed to have a bounded number of parameters, then the error bounds are meaningful when the amount of source data exceeds the number of translator parameters $|\Theta|$. For further intuition, consider any fixed translator with high error rate. Then a $1-\alpha$ fraction of its mistranslations will be implausible. These implausible translations will rule out a constant fraction of translators.
>
> As you noticed, the intuition in line 717 is incorrect, the $\lesssim$ and $\gtrsim$ have accidentally been swapped! So it should indeed be $A \wedge \neg B$ is unlikely. In the proof itself (line 747) we have them in the correct direction.
>
> > I would also suggest the authors to include some experiments...
>
> Done! See the pdf attached to our overall response.
>
> > I think there needs to be more justification for how the instantiations are related to real world scenarios...
>
> Definition 3.1, the Knowledge Graph, corresponds to an abstraction of the natural phenomenon of language composition. Real-world constraints arise due to the laws of nature that are common to humans and fish. Consider the fact that certain translations such as "The salmon ate the shark" are implausible while "The shark ate the salmon" is plausible (here the nodes correspond to different fish). Definition 3.1 captures binary relations such as this, and it only requires that the two graphs are correlated rather than fully aggree.
>
> > list assumptions more explicitly and clearly...
>
> The two assumptions underlying all our models, are:
>
> 1. There exists a hypothetical translator from the animal to human language (a hypothetical "mermaid") that does not have a prohibitively large number of parameters.
>
> For assumption 1, neural networks have proven useful for translation, e.g., between (human) languages, images and natural languages, speech and text, etc. It seems plausible that they could translate animal communication, or at least as well as a hypothetical mermaid could. As discussed in the submission, translators often require fewer parameters than the underlying LMs. Because human language is compositional, it is possible to translate the sentence "The tuna ate the mackerel" to French by knowing the meanings of tuna, mackerel and ate and their French equivalents (and grammar). In contrast, we have learned from GPTs that LMs can spit out encyclopedic articles about tuna and mackerel fish. Our knowledge graphs model this: e.g., for a “food web,” the nodes correspond to fish and the edges correspond to which fish eat which other fish. In some sense, the crux of the low-complexity translator assumption is that the source animal language is also compositional, deriving their meaning from the meaning of smaller units composed together.
>
> 2. There is common ground between the languages, e.g., there are certain things that are implausible in both languages.
>
> For assumption 2, LMs embed many properties of nature. If an animal language is sufficiently rich, it should also embed similar real-world knowledge. The common ground includes knowledge that "the shark ate the salmon" is more likely than "the salmon ate the shark".
>
> > Describe the meaning of the broader prior and agreement...
>
> The broad prior is over reasonable translations that we might see. Of course, one wouldn't be surprised to see translations of whale utterances to be "Let's dive and eat some squid." On the other hand, one would be shocked to learn that whales use smartphones, and many of their conversations are about these communications. See additional discussion in Appendix G.
>
> Agreement is easily illustrated in the food web example from before. Another way to think about it is, if we used a language model trained on animal communication to output detailed articles about all the fish, how many facts would be in common between that and the corresponding human-written (e.g., Wikipedia) articles.  That is what we refer to as agreement. Note that we do not require perfect agreement, as illustrated by the agreement parameter in our Knowledge Graph model and experiments.
>
> > they rarely provide explanations for the symbols or the formulas either... the symbols in Section 3.1 have different meanings than the ones in Section 3.2.
>
> Our two models are different but use related notation. We chose to use the similar symbols to capture the same concept in the two sections, but we understand that this can be confusing.  We will add tables of symbols to clarify their meanings.
>
> > In my opinion, this paper’s assumption about the existence of a set of plausible translations is very realistic and insightful for understanding the mechanism of unsupervised translation. As far as I know, attributing the success of unsupervised translation to this characteristic is novel and I think it is convincing (though may not be a complete explanation).
>
> > The knowledge graph model is inspiring to me. I think it suggests that the success of unsupervised translation is because of the interaction between some high-level structure and some low-level structure of language. In this model, the high-level structure is the plausibility of the combination of nodes, while the low-level structure is the one-to-one mapping of the nodes.
>
> Thank you for these positive remarks. Considering that our rebuttal clarifies and fixes the main technical issue you (astutely) observed, and our additional experiments, we humbly ask that you consider increasing your score.

---

> > ### Comment · Reviewer_FHLH · 2023-08-11
> >
> > Thank for your clarifications. After reading your experiment settings, I now understand the intuition of the proof.
> >
> > However, I start to wonder, isn't the assumption that $\theta^* \in \Theta$ a little bit like cheating? In your experiment, the ground truth translator is chosen based on the translator family. In this way you don't need to worry that the translator family is too small. But in the real world, we can only assume that a good enough $\theta$ is in $\Theta$ if our $|\Theta|$ is large enough. Therefore, if the error bound depends on $|\Theta|$, then it may be vacuous in practice. I think some more elaborations are required.

---

> > > ### Author Response · Authors · 2023-08-11
> > > **Realizability assumption**
> > >
> > > We again applaud your nuanced discussion and interest. This is a natural and standard question and we discuss it in Appendix I of the supplementary materials, specifically see Theorem I.1. We agree that this simplifying assumption is not realistic but it is very often used in the first model of a problem for the following reasons. Specifically, the assumption, often called the "Realizability assumption," that $\theta^* \in \Theta$ is completely standard within the field of Computational Learning Theory [1]. For example, in binary classification, the PAC learning model, for which Les Valiant was awarded the Turing award, makes the realizability assumption. It has been the main model of the field of Computational Learning Theory for decades and the realizability assumption is also considered heavily in the adjacent field of Statistical Learning Theory. A later generalization, called Agnostic learning, considers the general case where one drops this assumption and merely wishes to attain error near that of the best $\theta \in \Theta$. What has been found time and again across problems is that the data requirements of agnostic learning are polynomially related to those of PAC learning [2], i.e., PAC learning often requires $\tilde{O}(\epsilon^{-1})$ training data to achieve $\epsilon$ error while agnostic learning often requires $\tilde{O}(\epsilon^{-2})$ examples (the $\tilde{O}$ notation hides logarithmic factors) [see 1 and appendix I of our supp. material] . It would be relatively straightforward (but tedious) to extend our analysis to the agnostic setting, but we felt that as a first pass the simpler model was easier for readers to understand the theorem statements and proofs. We could do that analysis on one or all of the models if you would like. The MLE paradigm we analyze (used to train most LLMs) works well in practical language settings when the realizability assumption doesn't hold.
> > >
> > > # References
> > > [1] An Introduction to Computational Learning Theory. Michael J. Kearns, Umesh Vazirani. 1994 MIT Press
> > >
> > > [2] Realizable Learning is All You Need. Max Hopkins, Daniel Kane, Shachar Lovett, Gaurav Mahajan. COLT 2022.

---

> > > > ### Comment · Reviewer_FHLH · 2023-08-12
> > > >
> > > > Thank you for your reply.
> > > >
> > > > I think the main difference between this work and the PAC learning bound with the realizability assumption is that the proof of the PAC learning bound does not require any more assumptions on the hypothesis set than the realizability assumption. In contrast, in this work, Lemma C.1 is proven over the randomness of the problem distribution (i.e. $S$ is chosen randomly independently of $\theta$). It is an implicit assumption that ensures that any $\theta$ in the hypothesis is unlikely to be too misleading. I think that's why you can observe positive results in your toy experiment.
> > > >
> > > > I may be wrong, but this implicit assumption in Lemma C.1 seems to conflict with the presumption of Theorem A.2. In Theorem A.2, $\hat{\theta}$ is actually selected based on $S$, i.e. $\hat{\theta} = \mathrm{MLE}^{\rho}(x_1, \cdots, x_n)$. Therefore, the probability space of Theorem A.2 is different from the one of Lemma C.1 (given $\hat{\theta}$, $S$ is not totally random), so I don't think applying a union bound over Lemma C.1 and Theorem A.2 is reasonable, while this is how Theorem 3.4 is proven. Please let me know if I am wrong.
> > > >
> > > > Intuitively speaking, as long as in your hypothesis set you have more than one translators that don't generate nonsense, you can't have 0 error in expectation (because only one of them is perfect and the other ones are misleading). Of course you can argue that the probability that the hypothesis set contains a misleading translator is low when the hypothesis set is small and $|T|$ is large. But I would say, if $|T|$ is larger, meaning that the problem is harder, and your hypothesis set is small, then in practice, the realizability assumption is less likely to hold.

---

> > > > > ### Author Response · Authors · 2023-08-13
> > > > >
> > > > > Let us first explain the technical aspect (union bound in Theorem 3.4), and then give a high-level answer.
> > > > >
> > > > > **Union Bound in Theorem 3.4.**
> > > > > This is how we use Theorem A.2 and the union bound in the proof of Theorem 3.4. Lemma C.1 states that, just based on $S$ (regardless of the samples), $\epsilon_\gamma$  will be small with high probability. So there is no dependence on $\hat{\theta}$ or the particular samples $x_1, x_2, \ldots, x_m$ for this part. Once we have established that for most $S$'s, $\epsilon_\gamma$ is small, we can then use Theorem A.2 which says that, for those $\epsilon_\gamma$ which are small, with high probability the error of $\hat\theta$ will be small.
> > > > >
> > > > > Specifically, when we say "by the union bound", the two failure events we are considering are:
> > > > >
> > > > > 1. $Pr_S[\epsilon_\gamma > 6\gamma/\alpha] \le 2\gamma/3 $
> > > > > 2. $Pr_{S, x_1,\ldots x_m}[\mathcal{L}(\hat{\theta}) > \epsilon_\gamma] \le \gamma/3$
> > > > >
> > > > > Two events do not have to be independent in order to apply the union bound, and in order for $\mathcal{L}(\hat\theta)>6 \gamma/\alpha$, one of those two failures events must occur. Lemma C.1 bounds the failure probability of the first event. To see how Theorem A.2 bounds the probability this latter event (the crux of the discussion), note that fixing $S$ implies fixing $\tau,\rho$ in which case $\epsilon_\gamma$ is determined. Let's write it as $\epsilon_\gamma^{S}$ to make the dependence on $S$ explicit. The second event is:
> > > > >
> > > > > $$Pr_{S, x_1,\ldots, x_m}\left[ ~\mathcal{L}(\hat{\theta}) > \epsilon_\gamma^S \right] = \sum_{S\subseteq \mathcal{Y}} Pr[S] \cdot Pr_{x_1,\ldots,x_m}\left[ ~\mathcal{L}(\hat{\theta}) > \epsilon^S_\gamma ~ | ~  S \right].$$
> > > > >
> > > > > In turn, you can see how the right hand side above has $S$ fixed, because we have conditioned on it, and thus by Theorem A.2 it is at most $\delta/3$ for **each and every** $S\subseteq \mathcal{Y}$. Thus the right hand side above is,
> > > > > $$\sum_S Pr[S] \cdot \delta/3 = \delta/3,$$
> > > > > because $\sum_S Pr[S] =1$.
> > > > >
> > > > > **High-level.**
> > > > > At a high level, we can decompose the technical contributions of our paper into three parts:
> > > > >
> > > > > 1. Formalizing the problem of UMT with a prior.
> > > > > 2. Proving a general theorem relating divergence and loss via plausible ambiguities (Theorem A.2).
> > > > > 3. Proposing two stochastic, parameterized models of language, and analyzing in which parameter regimes the sampled languages are translatable.
> > > > >
> > > > > We would like to emphasize that Part 2 (the general theorem, which in this discussion thread has been compared to realizable PAC learning bounds) is independent of the stochastic models of language. Separately, we analyze the stylized models of Part 3 to demonstrate how language complexity and common ground affect the sample complexity of unsupervised translation. As these are stochastic models, translatability is actually "the sampled languages are translatable with high probability." In the Common Nonsense experiments, $S$ and $\Theta$ are chosen randomly and independently, so we do not see how this "ensures that any $\theta$ is not too misleading." The positive results observed occur only for sufficiently many samples (as predicted, more samples are needed for smaller $\alpha$; indeed, for insufficient samples, the learned translator has very high error---please let us know if we understood your comment on the experiments correctly.
> > > > >
> > > > > Thanks again for your interest attention to detail!

---

> > > > > > ### Comment · Reviewer_FHLH · 2023-08-14
> > > > > >
> > > > > > Thank you for your elaboration. After rethinking the proof, I think the technical part is correct. Sorry for taking your time.
> > > > > >
> > > > > > ## High-level
> > > > > >
> > > > > > Let me explain what I meant by "... that ensures that any in the hypothesis is unlikely to be too misleading" first.
> > > > > > I meant, if we select $\Theta$ without any knowledge about $S$, then it's unlikely any $\theta$ that has low $\hat{v}(\theta)$ will be selected,
> > > > > > i.e. $\lvert A_\gamma  \backslash  \\{ \theta^* \\} \rvert$ is likely to be 0.
> > > > > > Note that $A_\gamma  \backslash  \\{ \theta^* \\}$ is the set of $\theta$ that have misleadingly high plausibility $\hat{v}(\theta)$ but have high loss $L$.
> > > > > > If you have the realizability assumption and have a small $\lvert A_\gamma \rvert$ then surely you are very likely to find $\theta^*$ by MLE, because it is very likely that $\theta^*$ is the only element in $A_\gamma$.
> > > > > >
> > > > > > So I think the realizability assumption is there not just for a simpler analysis. It's actually essential for a non-trivial bound. I think if you remove this assumption from your toy experiment, you won't be able to see the same positive results. I have mentioned my intuition before: MLE can not help you distinguish which translator in an ambiguous set $A$ has a lower error rate, no matter how many samples  you have (i.e. how large $m$ is). I think the expected accuracy you can get is at most $\frac{1}{|T|}$ if you remove the the realizability assumption. Please correct me if you actually are able to get positive results without the realizability assumption.
> > > > > >
> > > > > > (Running a toy experiment for this is not infeasible because you can choose a much smaller $|T|$ and a large enough $|\Theta|$ that ensure $\theta^*$ has a high probability to be included in $\Theta$.)
> > > > > >
> > > > > > Some observations we can have from the technical results can make my point above clearer.
> > > > > > - Lemma C.1 is not about sample complexity. The bound is even better when the problem is more difficult, i.e. $|T|$ is larger. I think it is because when $|T|$ is larger, it is more likely that $A_\gamma$ is small.
> > > > > > - The error bound of Theorem 3.4 does not converge to 0 with infinite samples, i.e. $m \to \infty$. It is aligned with my intuition.

---

> > > > > > > ### Author Response · Authors · 2023-08-14
> > > > > > >
> > > > > > > Thank you for confirming the correctness of the proof. We are grateful for *your* time!
> > > > > > >
> > > > > > > **On the role of the realizability assumption.** We are happy to conduct an experiment to demonstrate robustness of our claim to noise, i.e., that it generally holds when the realizability assumption is relaxed. Would this address your main remaining reservation?

---

> > > > > > > > ### Comment · Reviewer_FHLH · 2023-08-17
> > > > > > > >
> > > > > > > > Sorry for the late reply. I ran a quick experiment with $|\Theta| = 10000, |P| = 20, |T|=5, \alpha=0.75$ with an oracle for $\hat{v}(\theta)$. The average accuracy I got is 17.56%. When $|T| = 4$, the accuracy is 23.675. So the accuracies of the two experiments are less than $1/|T|$, just as I said.
> > > > > > > >
> > > > > > > > Please let me know if I misunderstood something.
> > > > > > > >
> > > > > > > > ```python
> > > > > > > > import numpy as np
> > > > > > > >
> > > > > > > >
> > > > > > > > T = 5
> > > > > > > > P = 20
> > > > > > > > alpha = 0.75
> > > > > > > >
> > > > > > > > assert 1 - alpha >= T / P
> > > > > > > >
> > > > > > > > n = 10000
> > > > > > > > accs = []
> > > > > > > > for t in range(1000):
> > > > > > > >     rng = np.random.default_rng(42 + t)
> > > > > > > >     fs = [
> > > > > > > >         rng.choice(P, T, replace=False)
> > > > > > > >         for _ in range(n)
> > > > > > > >     ]
> > > > > > > >
> > > > > > > >     # assume we have an oracle for the likelihood
> > > > > > > >     ll = np.array([
> > > > > > > >         (f < P * (1 - alpha)).mean()
> > > > > > > >         for f in fs
> > > > > > > >     ])
> > > > > > > >     best = np.argmax(ll)
> > > > > > > >
> > > > > > > >     # target function is always np.arange(T)
> > > > > > > >     acc = np.mean(fs[best] == np.arange(T))
> > > > > > > >     accs.append(acc)
> > > > > > > >
> > > > > > > > print('mean acc', np.mean(accs))
> > > > > > > > ```
> > > > > > > >
> > > > > > > > On the other hand, removing the realizability assumption may not be that much a problem for the Knowledge Graph model, I guess.
> > > > > > > > If it is really the case, I think the idea discussed in this paper is still inspiring.
> > > > > > > > I would suggest the authors to make this paper more accessible to have a greater impact.
> > > > > > > > This may include:
> > > > > > > >
> > > > > > > > 1. Stating the assumptions more clearly and justifying them well, i.e. how are the assumptions connected to the real-world scenario.
> > > > > > > > 2. Starting with a simpler setup. You don't need to start with a complicated setup though it may be closer to the real-world scenario. For example, I think some results will be easier to understand if we don't take sample complexity into account at the beginning (assuming that we have an oracle for $\hat{v}(\theta)$.
> > > > > > > > 3. Providing some intuitive toy thought experiments.
> > > > > > > > 4. Providing high-level proof ideas in the main text. Personally I really look forward to a polished version of this work and I hope I don't need to spend many hours understanding a proof.
> > > > > > > >
> > > > > > > > Thank you again for the clarifications you made.

---

> > > > > > > > > ### Author Response · Authors · 2023-08-21
> > > > > > > > > **Agnostic model and experiment**
> > > > > > > > >
> > > > > > > > > Thank you for deepening the discussion.
> > > > > > > > >
> > > > > > > > > We would do the agnostic experiment differently. In particular, if we understand your implementation, you had no "good" translator to begin with, whereas in our agnostic model we
> > > > > > > > > would have a translator with some given error rate, say $\eta$. This relaxes the realizable assumption of a "0-error mermaid translator" rather than removing it entirely. In the experiments below, we have planted a mermaid translator with error
> > > > > > > > > $\eta \approx 0.2$. As you can see, as the language gets more complex, our accuracy approaches that of the best translator. However, for languages that are simple our error is quite high,
> > > > > > > > > further corroborating the main messages in the paper.
> > > > > > > > >
> > > > > > > > > |  P  |  MLE error  |  best translator error |
> > > > > > > > > | --- | ----------  | ---------------------- |
> > > > > > > > > |  32 |        0.93 |                   0.16 |
> > > > > > > > > |  64 |        0.94 |                   0.20 |
> > > > > > > > > | 128 |        0.67 |                   0.20 |
> > > > > > > > > | 256 |        0.26 |                   0.20 |
> > > > > > > > > | 512 |        0.20 |                   0.20 |
> > > > > > > > > |1024 |        0.20 |                   0.20 |
> > > > > > > > >
> > > > > > > > > Notice that the "regret", the difference between our MLE error and the best error in the class shrinks as the language size P grows.
> > > > > > > > > Further details can be found in the code at the end of this message.
> > > > > > > > >
> > > > > > > > > **Thank you.** Over the past dozen messages, we have had a productive discussion about the technical details of our proof, the
> > > > > > > > > intuition underlying our theory, its empirical validation, and the possibility of extending our theory to the agnostic setting.
> > > > > > > > >
> > > > > > > > > In particular, you had doubts about the correctness of Line 714 in our proof, and the correctness of the application
> > > > > > > > > of the union bound in the proof of Theorem 3.4. The clarifications we will make will may help other readers as well.
> > > > > > > > > Once again, we thank you for your attention to detail, and for verifying that our proof is correct.
> > > > > > > > >
> > > > > > > > > At your suggestion, we have also devised experiments to verify our theory. In your words,
> > > > > > > > > > After reading your experiment settings, I now understand the intuition of the proof.
> > > > > > > > >
> > > > > > > > > It was pleasing to see our experiments empirically validate our theory, but it was even more satisfying for us to hear
> > > > > > > > > that they also helped illuminate the proof. Coming full circle, we even used the new experiment (fruit of our discussion)
> > > > > > > > > as empirical evidence that our theory could (as expected) be extended to the agnostic case.
> > > > > > > > >
> > > > > > > > > To conclude, our paper has improved from this discussion, in which we addressed your concerns. Given these improvements,
> > > > > > > > > we ask that you consider raising your score.
> > > > > > > > >
> > > > > > > > > # Code
> > > > > > > > >
> > > > > > > > > ```python
> > > > > > > > > import numpy as np
> > > > > > > > >
> > > > > > > > > NUM_TRANSLATORS = 10_000
> > > > > > > > > NUM_RUNS = 10_000
> > > > > > > > > MERMAID_ERROR = 0.2
> > > > > > > > > FRAC_COMMON_NONSENSE = 0.5
> > > > > > > > > LOW_PROB = 0.01  # agnostic prior cannot assign zero probability to nonsense
> > > > > > > > > PS = [32, 64, 128, 256, 512, 1024]
> > > > > > > > >
> > > > > > > > >
> > > > > > > > > def experiment(p, t, alpha, seed, mermaid_error, num_translators, low_prob):
> > > > > > > > >     assert 1 <= t <= p
> > > > > > > > >
> > > > > > > > >     rng = np.random.default_rng(seed)
> > > > > > > > >
> > > > > > > > >     ground_truth_translator = np.arange(p)
> > > > > > > > >     mermaid = np.copy(ground_truth_translator)
> > > > > > > > >
> > > > > > > > >     while np.mean(mermaid != ground_truth_translator) < mermaid_error:
> > > > > > > > >         # swap two random elements
> > > > > > > > >         i, j = rng.choice(t, 2, replace=False)
> > > > > > > > >         mermaid[i], mermaid[j] = mermaid[j], mermaid[i]
> > > > > > > > >
> > > > > > > > >     translators = [mermaid] + [rng.permutation(p) for _ in range(num_translators - 1)]
> > > > > > > > >     rng.shuffle(translators)  # hide the mermaid among others
> > > > > > > > >
> > > > > > > > >     common_nonsense = rng.choice(range(p), int(alpha * p), replace=False)
> > > > > > > > >
> > > > > > > > >     rho = np.ones(p)  # the prior
> > > > > > > > >     rho[common_nonsense] = low_prob
> > > > > > > > >     rho /= rho.sum()  # normalize to sum to 1
> > > > > > > > >
> > > > > > > > >     picked_tau = False
> > > > > > > > >     while not picked_tau:
> > > > > > > > >         tau = np.zeros(p)
> > > > > > > > >         tau[rng.choice(range(p), t, replace=False)] = 1.0
> > > > > > > > >         tau[common_nonsense] = 0.0
> > > > > > > > >         picked_tau = sum(tau) > 0.0
> > > > > > > > >
> > > > > > > > >     tau /= tau.sum()  # normalize to sum to 1
> > > > > > > > >
> > > > > > > > >     def raw_error(tau, translator):
> > > > > > > > >         return np.sum(tau * [i != j for i, j in enumerate(translator)])
> > > > > > > > >
> > > > > > > > >     def log_likelihood(rho, tau, translator):  # assume infinite m for now
> > > > > > > > >         log_probs = np.log([rho[i] for i in translator])
> > > > > > > > >         return np.sum(tau * log_probs)
> > > > > > > > >
> > > > > > > > >     learned_translator = max(translators, key=lambda t: log_likelihood(rho, tau, t))
> > > > > > > > >     best_translator = min(translators, key=lambda t: raw_error(tau, t))
> > > > > > > > >     return raw_error(tau, learned_translator), raw_error(tau, best_translator)
> > > > > > > > >
> > > > > > > > > print("|  P  |  MLE error  |  best translator error |")
> > > > > > > > > print("| --- | ----------  | ---------------------- |")
> > > > > > > > > for p in PS:
> > > > > > > > >     t = p // 4  # always have |T| = |P|/4 for simplicity
> > > > > > > > >     res = [
> > > > > > > > >         experiment(p, t, FRAC_COMMON_NONSENSE, seed, MERMAID_ERROR, NUM_TRANSLATORS, LOW_PROB)
> > > > > > > > >         for seed in range(NUM_RUNS)
> > > > > > > > >     ]
> > > > > > > > >     raw, best = np.mean(res, axis=0)
> > > > > > > > >     print(f"| {p:3} | {raw:11.2f} | {best:22.2f} |")
> > > > > > > > >
> > > > > > > > > ```

---

### Official Review · Reviewer_CDK4 · 2023-06-26

**Soundness:** 4 excellent
**Presentation:** 2 fair
**Contribution:** 4 excellent
**Rating:** 8
**Confidence:** 4

**Summary:**

### After rebuttal

Thanks the authors for the detailed rebuttal, especially with new experiments. Provided the condition that the authors will add the new experiments and adjust the presentation after reviewers' comments, I gladly change my scores and recommendation!

----
The paper presents a rather interesting theory about how unsupervised machine translation (UMT) without any parallel data. The paper taps into the analysis of world knowledge graph models, which may model relationships between objects/subjects of the world, and presented by words in the data. And suggests that it is the compositionality of languages somewhat allows UMT to work.

The paper also analyze the common nonsense model concept, where UMT generally tries to rotate the source distribution so that it can fit in a unique orientation in the target distribution, given enough nonsense data.

The paper presents some conclusion about the theorical error rate of UMT in relation with data and languages, though no experiments were done.

**Strengths:**

* The concept and analysis of the UMT in the paper is very interesting and novel.
* The theoretical models of random knowledge graph and common nonsense models are interesting and should be investigated further.
* Some observation from such formulation is that error rate is inversely related with language complexity, which is indeed non-intuitive, but there is no experiment to confirm such observation.

**Weaknesses:**


* The paper is very math heavy with limited visualization or intuitive explanation. This makes it rather difficult to comprehend and and grasp the content. More intuitive explanation with figures and illustration are needed to understand. I am also unsure about the correctness of the formulation and the math, and can only assume it to be correct in the review.
* There is no experiments to validate the theory at all, even with a toy experiments. This makes it hard to conclude the completeness of the paper.
* The organization is not very well-structured and rather strange. There are no experiments and related work sections.
* Despite heavily advertised in the initial part of the paper, there is limited correlation, conclusion, or elaboration about how we can model such human-animal communications, which is quite disappointing.
* There is no clear indication and suggestion of what to do with UMT after reading this theory. The paper does not make recommendations on how to design better UMT models, how to train them, which aspects of existing UMT models need to be fixed so that it can handle low-resource languages and more ambitiously human-animal translations. This is real challenges that UMT performs very poorly when domains of unlabeled data sources are very distant.

Given the high degree of novelty of implication of this work, I would give much higher score if the rebuttal addresses some of the concerns.

**Questions:**

N.A

**Limitations:**

There is no discussion of limitations.

---

> ### Author Rebuttal · Authors · 2023-08-10
>
> > There is no experiments to validate the theory at all, even with a toy experiments.
>
> Thank you for this great suggestion. To address this, we have added synthetic experiments to validate our theoretical results. See the pdf attached to our overall response.
>
> > The paper is very math heavy with limited visualization or intuitive explanation.
>
> We have dedicated a significant amount of space (Figures 1-3 in the original submission) to visualizations aiming to provide intuition. We hope the newly-added experiments provide even more intuition.
>
> > The organization is not very well-structured and rather strange.
>
> We will make our assumptions (and justification) more explicit as follows:
>
> 1. There exists a hypothetical translator from the animal to human language (e.g., a hypothetical "mermaid" Transformer) that does not have a prohibitively large number of parameters, specifically the number of parameters needs to be fewer than the number of animal data.
>
> For assumption 1, neural networks have proven useful for translation, e.g., between (human) natural languages, between images and natural languages, speech and text, between programming and natural languages, etc. It seems plausible that they could translate animal communication, or at least as well as a hypothetical mermaid could if one existed. To understand the number of parameters required, as discussed in the submission, translators often require fewer parameters than the underlying LMs. Because human language is compositional, it is possible to translate the sentence "The tuna ate the mackerel" to French by knowing the meanings of tuna, mackerel and ate and their French equivalents (and grammar)--one does not need to apply the knowledge of whether tuna eat mackerel or vice versa. In contrast, we have learned from GPTs that a good LM must be able to spit out encyclopedic articles about tuna and mackerel fish. Our knowledge graphs model this type of phenomenon, for example for a “food web,” the nodes correspond to fish and the edges correspond to which fish eat which other fish. In some sense, the crux of the low-complexity translator assumption is that the source animal language is also compositional, deriving their meaning from the meaning of smaller units composed together. As discussed in the paper, some simple animal languages have already been shown to be compositional, and compositionality is useful for language efficiency.
>
> 2. There is common ground between the languages, e.g., there are certain things that are implausible in both languages.
>
> For assumption 2, LMs embed many properties of nature, as can be seen by the fact that GPTs (trained unsupervised on natural language corpora) can write accurate articles about each fish, currents, diving, and many other topics, full of details that may also be known to animals such as whales. If an animal language is sufficiently rich, it should also embed similar real-world knowledge. The common ground includes knowledge that "the shark ate the salmon" is more likely than "the salmon ate the shark".
>
> Moreover, as discussed in the paper, there are currently several major interdisciplinary efforts under way to translate animal communication  [1, 2] involving interdisciplinary teams of linguists, marine biologists, and computer scientists, whose efforts are resting on the assumption that translation is possible. Our models formalize what assumptions are required for unsupervised translation.
>
>
> > There is no clear indication and suggestion of what to do with UMT after reading this theory. The paper does not make recommendations on how to design better UMT models, how to train them, which aspects of existing UMT models need to be fixed so that it can handle low-resource languages and more ambitiously human-animal translations.
>
> While we agree algorithms are important, theoretical ML results often provide insights rather than new algorithms, such as "The no free-lunch theorem". To quote another reviewer, *"These findings may have implications for the quantity and type of communication data that is collected for deciphering animal communication and for UMT more generally."*
>
> In particular, a fundamental question facing scientists working on translating animal communication is: **how much and what kinds of data need to be collected?** Our findings suggest that unsupervised data alone may suffice, at least from an information-theoretic/sample-complexity perspective. For example, [1] discusses that the project must decide how to allocate its data-collection investments between pure audio and other more expensive modalities such as video. Our work may shed light in this direction.
>
> And to quote another reviewer discussing their takeaways regarding how UMT works: *In my opinion, this paper’s assumption about the existence of a set of plausible translations is very realistic and insightful for understanding the mechanism of unsupervised translation. As far as I know, attributing the success of unsupervised translation to this characteristic is novel and I think it is convincing (though may not be a complete explanation).* They then go on to say,
> *"The knowledge graph model is inspiring to me. I think it suggests that the success of unsupervised translation is because of the interaction between some high-level structure and some low-level structure of language. In this model, the high-level structure is the plausibility of the combination of nodes, while the low-level structure is the one-to-one mapping of the nodes."*
>
>
> > There are no experiments and related work sections.
>
> Thank you for the suggestion. If our paper is accepted, we will be given an extra page and will add a related work section (currently in Appendix E) to the paper. As mentioned, we have added synthetic experiments to validate our theoretical results.
>
> ## References
> [1] Jacob Andreas et al., Toward understanding the communication in sperm whales. iScience, 2022.
>
> [2] Emily Anthes. The animal translators. The New York Times, 2022.

---

### Official Review · Reviewer_LYrz · 2023-07-02

**Soundness:** 3 good
**Presentation:** 2 fair
**Contribution:** 3 good
**Rating:** 6
**Confidence:** 2

**Summary:**

This paper introduces a theoretical framework for unsupervised machine translation (UMT) when no parallel data available and source and target language do not share similar linguistic structures. To do this, they first define and clarify the three main challenges: understanding the goal, no linguistic structure shared and domain gap. In response to these challenges, they propose a general framework and instantiate it with two models: knowledge graph model , which is highly structured with text representing an edge between a pair of nodes, and "Common nonsense" model which is completely unstructured. For both models, they establish the error bounds and demonstrate that the error rate is inversely related to the amount of samples, common ground, and the language complexity.

**Strengths:**

* The paper presents a novel theoretical framework for UMT, which is needed but challenging in the field of UMT.
* The authors propose two complementary models of language, one that is highly structured and one that is completely unstructured, which provides a relatively comprehensive analysis of UMT.
* The authors provide theoretical bounds on the data requirements for UMT, which can inform how much data are collected for such tasks.
* The authors' ideas are interesting and insightful, such as the motivating example in the knowledge graph model where one can complete the mapping based on the similar structure of the graph, and enough "nonsense" make a nearly unique mapping in "common nonsense" model.

**Weaknesses:**

* The paper makes strong assumptions about the language, which may limit the applicability of the proposed models in real-world scenarios.
* The paper does not provide any empirical validation of the proposed theoretical framework and models. The absence of experimental results makes it difficult to assess the practical effectiveness of the proposed methods.
* The paper does not provide a clear explanation of how the proposed models can be implemented in practice, which may limit their usability for practitioners in the field.

**Questions:**

* I am not sure what the "prior" proposed in the text refers to, but I tentatively comprehend it to be a prefix LM, where the prefix may be a prompt like "a whale said:".
* Why maximizing the likelihood of the prior implies translation. After all, the source language is not involved in the calculation of this likelihood, so how to reflect the fidelity of the translation?

**Limitations:**

Same as Weaknesses. Or can we consider a real experiment, such as mutual translation between knowledge graphs constructed by English and Chinese respectively?

---

> ### Author Rebuttal · Authors · 2023-08-10
>
> Thank you for your review and feedback.
>
> > The paper does not provide any empirical validation of the proposed theoretical framework and models.
>
> Thank you for this great suggestion. To address this, we have added synthetic experiments to validate our theoretical results. See the pdf attached to our overall response.
>
> > The paper does not provide a clear explanation of how the proposed models can be implemented in practice, which may limit their usability for practitioners in the field.
>
> We believe strongly that our theoretical findings are illuminating despite the fact that we do not introduce a new algorithm (theoretical ML results often provide insights rather than new algorithms, such as *The no free-lunch theorem* and *The lottery ticket hypothesis*). To quote the reviewer who gave us a score of 9 :-), *"These findings may have implications for the quantity and type of communication data that is collected for deciphering animal communication and for UMT more generally."*
>
> In particular, a fundamental question facing scientists working on translating animal communication is: **how much and what kinds of data need to be collected?** Our findings suggest that unsupervised data alone may suffice, at least from an information-theoretic/sample-complexity perspective. For example, [1] discusses that the project must decide how to allocate its data-collection investments between pure audio and other more expensive modalities such as video. Our work may shed light in this direction.
>
> And to quote another reviewer discussing their takeaways regarding how UMT works: *In my opinion, this paper’s assumption about the existence of a set of plausible translations is very realistic and insightful for understanding the mechanism of unsupervised translation. As far as I know, attributing the success of unsupervised translation to this characteristic is novel and I think it is convincing (though may not be a complete explanation).* They then go on to say,
> *"The knowledge graph model is inspiring to me. I think it suggests that the success of unsupervised translation is because of the interaction between some high-level structure and some low-level structure of language. In this model, the high-level structure is the plausibility of the combination of nodes, while the low-level structure is the one-to-one mapping of the nodes."*
>
>
> > I am not sure what the "prior" proposed in the text refers to, but I tentatively comprehend it to be a prefix LM, where the prefix may be a prompt like "a whale said:".
>
> Yes, that is correct. The prior is a probabilistic model of text (i.e., a LM) which we anticipate might be generated by the translator.
>
> > Why maximizing the likelihood of the prior implies translation. After all, the source language is not involved in the calculation of this likelihood, so how to reflect the fidelity of the translation?
>
> We are choosing the translator parameters (not the prior, which remains fixed) so as to maximize the probability (under the fixed prior) of the translations of the source data. In this way, the source language samples are crucially involved in the optimization process.

---

> > ### Comment · Reviewer_LYrz · 2023-08-18
> >
> > Acknowledging that I've read the response.
> >
> > ---
> >
> > After reading the author's responses and discussions with other reviewers, I would like to see this paper accepted.

---

### Official Review · Reviewer_7v3M · 2023-07-07

**Soundness:** 2 fair
**Presentation:** 2 fair
**Contribution:** 2 fair
**Rating:** 4
**Confidence:** 2

**Summary:**

The paper describes a theoretical analysis of error bounds for unsupervised machine translation, motivated by considering the unusual, and so far only speculative, case of translation between animal "languages" to human natural languages.

The paper consider the case where a number of examples of the source (animal) language utterances are available, as well as a prior on the target language (e.g., a conventional pretrained LLM), and attempts to investigate what error an invertible translation function trained to maximize a MLE objective may achieve compared to a hypothetical golden truth translator.

The paper considers two specific cases with different assumptions over the languages and the priors: in the first case the languages are assumed to be Erdős–Rényi random graphs parametrized by an edge agreement parameter. The authors prove a bound which does not vanish even in the limit of infinite source sentences but vanishes in the limit of perfect agreement. I do not find this model to be very realistic for natural languages.

The second case consider the two languages to be uniform distributions over a subset that excludes a "nonsense" set. The authors prove an bound similar to supervised learning. I am not sure why uniformity is considered a plausible hypothesis here, as actual natural languages are characterized by power law distributions.

Overall, I think there is not enough justification connecting the formal models with the motivation of translating animal communication.


**Strengths:**

- Novel and interesting topic.

- Theoretical analysis.

**Weaknesses:**

- Assumptions not well justified

- Formal model not well connected with motivation

- Too much background and related work is missing. Some is provided in the supplementary material, but it should be in the main paper.

**Questions:**

Is there any evidence that the formal models you propose can correctly capture the relevant factors of unsupervised translation even between human languages, let alone between animal and human languages?



**Limitations:**

Yes.

---

> ### Author Rebuttal · Authors · 2023-08-10
>
> Thank you for your review and feedback. We are glad you find the topic novel and interesting.
>
> > Assumptions not well justified
>
> We make the assumptions and justification more explicit in the paper, as follows. The two assumptions underlying all our models, are:
>
> 1. There exists a hypothetical translator from the animal to human language (e.g., a hypothetical "mermaid" Transformer) that does not have a prohibitively large number of parameters, specifically the number of parameters needs to be fewer than the number of animal data.
>
> For assumption 1, neural networks have proven useful for translation, e.g., between (human) natural languages, between images and natural languages, speech and text, between programming and natural languages, etc. It seems plausible that they could translate animal communication, or at least as well as a hypothetical mermaid could if one existed. To understand the number of parameters required, as discussed in the submission, translators often require fewer parameters than the underlying LMs. Because human language is compositional, it is possible to translate the sentence "The tuna ate the mackerel" to French by knowing the meanings of tuna, mackerel and ate and their French equivalents (and grammar)--one does not need to apply the knowledge of whether tuna eat mackerel or vice versa. In contrast, we have learned from GPTs that a good LM must be able to spit out encyclopedic articles about tuna and mackerel fish. Our knowledge graphs model this type of phenomenon, for example for a “food web,” the nodes correspond to fish and the edges correspond to which fish eat which other fish. In some sense, the crux of the low-complexity translator assumption is that the source animal language is also compositional, deriving their meaning from the meaning of smaller units composed together. As discussed in the paper, some simple animal languages have already been shown to be compositional, and compositionality is useful for language efficiency.
>
> 2. There is common ground between the languages, e.g., there are certain things that are implausible in both languages.
>
> For assumption 2, LMs embed many properties of nature, as can be seen by the fact that GPTs (trained unsupervised on natural language corpora) can write accurate articles about each fish, currents, diving, and many other topics, full of details that may also be known to animals such as whales. If an animal language is sufficiently rich, it should also embed similar real-world knowledge. The common ground includes knowledge that "the shark ate the salmon" is more likely than "the salmon ate the shark".
>
> Moreover, as discussed in the paper, there are currently several major interdisciplinary efforts under way to translate animal communication  [1, 2] involving interdisciplinary teams of linguists, marine biologists, and computer scientists, whose efforts are resting on the assumption that translation is possible. Our models formalize what assumptions are required for unsupervised translation.
>
> > Too much background and related work is missing. Some is provided in the supplementary material, but it should be in the main paper.
>
> Thank you for the suggestion. If our paper is accepted, we will be given an extra page and will add a related work section (currently in Appendix E) to the paper.
>
> > Is there any evidence that the formal models you propose can correctly capture the relevant factors of unsupervised translation even between human languages, let alone between animal and human languages?
>
> For unsupervised translation between human languages, the main findings are consistent with our models: more data and more common ground improves translation accuracy. For example, accuracy increases as the amount of ``common ground'' or overlap in training data topics [3].
>
>
> ## References
>
> [1] Jacob Andreas, (and 19 others). Toward understanding the communication in sperm whales. iScience, 2022.
>
> [2] Emily Anthes. The animal translators. The New York Times, 2022.
>
> [3] Jiajun Shen, Peng-Jen Chen, Matt Le, Junxian He, Jiatao Gu, Myle Ott, Michael Auli, Marc'Aurelio Ranzato. The Source-Target Domain Mismatch Problem in Machine Translation. EACL, 2021.

---

### Official Review · Reviewer_5Kio · 2023-07-28

**Soundness:** 2 fair
**Presentation:** 3 good
**Contribution:** 2 fair
**Rating:** 6
**Confidence:** 3

**Summary:**

### Update

I'm satisfied with the author's rebuttal and decided to overall improve my score.


This paper proposes a theoretical framework for analyzing Unsupervised MT motivated by animal communication. The paper is only the first step in the theoretical analysis of UMT. The theoretical framework was analyzed for two models: Knowledge graph and Common nonsense models. The authors show theoretically that the error rate is inversely related to the number of samples, common ground, and language complexity.

**Strengths:**

The authors work on providing a theoretical framework for analyzing unsupervised MT. The task is motivated by a real-world challenge of understanding animal communication.

The paper is well-written and was not too hard to follow. The general framework was well-defined and motivated.

The paper instantiated two models with common-nonsense and knowledge graphs. The paper derived theoretical bounds for error rates and provided comprehensive proof.

**Weaknesses:**

Purely descriptive: what is the key takeaway from the paper? How can we improve unsupervised MT based on these findings? Where can we use the theoretical framework?

No experimental results? I realize the paper focuses on the theory of unsupervised translation. However, none of the author's theoretical claims are validated using experimental findings which I find a bit hard to follow.

The authors claim that the error rate is inversely proportional to language complexity. This is counterintuitive to what I would expect in Machine Translation. However, there is no in-depth discussion, reasoning, or intuition as to why this is the case. Additionally, the related work or motivation section is missing in the main paper, where I believe they should motivate UMT.

I missed the connection in the paper with learning animal communication. The paper initially has a strong motivation to learn animal communication, however, the discussion section is weak and I could not find what to conclude, or whether can we actually use the unsupervised MT theoretical framework.

**Questions:**

Could the authors provide experimental findings for their theoretical framework? Take a simple case of language translation example and validate their theoretical findings using a few experiments. Right now, the paper focuses on the theoretical framework only.

Provide more intuition on why they found error rates are inversely proportional to language complexity.

Improve the connection with learning animal communication. There was very little discussion about animal communication apart from the introduction section.

**Limitations:**

The authors did not provide any limitations on their work.

---

> ### Author Rebuttal · Authors · 2023-08-10
>
> Thank you for your review and detailed feedback.
>
> Weaknesses:
> > Purely descriptive: what is the key takeaway from the paper? How can we improve unsupervised MT based on these findings? Where can we use the theoretical framework?
>
> We believe strongly that our theoretical findings are illuminating despite the fact that we do not introduce a new algorithm (theoretical ML results often provide insights rather than new algorithms, such as *The no free-lunch theorem* and *The lottery ticket hypothesis*). To quote the reviewer who gave us a score of 9 :-), *"These findings may have implications for the quantity and type of communication data that is collected for deciphering animal communication and for UMT more generally."*
>
> In particular, a fundamental question facing scientists working on translating animal communication is: **how much and what kinds of data need to be collected?** Our findings suggest that unsupervised data alone may suffice, at least from an information-theoretic/sample-complexity perspective. For example, [1] discusses that the project must decide how to allocate its data-collection investments between pure audio and other more expensive modalities such as video. Our work may shed light in this direction.
>
> And to quote another reviewer discussing their takeaways regarding how UMT works: *In my opinion, this paper’s assumption about the existence of a set of plausible translations is very realistic and insightful for understanding the mechanism of unsupervised translation. As far as I know, attributing the success of unsupervised translation to this characteristic is novel and I think it is convincing (though may not be a complete explanation).* They then go on to say,
> *"The knowledge graph model is inspiring to me. I think it suggests that the success of unsupervised translation is because of the interaction between some high-level structure and some low-level structure of language. In this model, the high-level structure is the plausibility of the combination of nodes, while the low-level structure is the one-to-one mapping of the nodes."*
>
> > No experimental results? I realize the paper focuses on the theory of unsupervised translation. However, none of the author's theoretical claims are validated using experimental findings which I find a bit hard to follow.
>
> Thank you for this great suggestion. To address this, we have added synthetic experiments to validate our theoretical results. **See the new experiments in the pdf attached to our overall response.**
>
> > The authors claim that the error rate is inversely proportional to language complexity. This is counterintuitive to what I would expect in Machine Translation. However, there is no in-depth discussion, reasoning, or intuition as to why this is the case.
>
> We are glad you find the result surprising. The intuition is that simple languages, as we see in our experiments, admit many plausible (but erroneous) translations. Consider a trivial language (e.g., of a cow) which has two utterances, $A$ and $B$, meaning "I'm happy" or "I'm sad"--one could not confidently get error less than 50\% in an unsupervised setting without any clue about which is A and which is B.  On the other hand, real-world constraints mean that we expect translations like "The shark ate the salmon" more than "The salmon ate the shark," which can make it feasible to eliminate a constant fraction of incorrect translators. More complex languages have more such constraints that rule out more erroneous translators. An example is illustrated in Figure 2 in the original submission, and we will add more intuition to the paper. In analogy, imagine solving a hard jigsaw puzzle with only blank white pieces. If there were just two identical square pieces, it would be trivial to fit them together but there would be no way to gain confidence that you had the pieces correctly positioned. In contrast, if all the pieces had unique, complex shapes, there may be only one way to fit them all together.
>
> > Additionally, the related work or motivation section is missing in the main paper, where I believe they should motivate UMT.
>
> Thank you for the suggestion. If our paper is accepted, we will be given an extra page and will add a related work section (currently in Appendix E) to the paper. The main motivation for the work is understanding animal communication, as discussed in the introduction.
>
> # References
>
> [1] Jacob Andreas, (and 19 others). Toward understanding the communication in sperm whales. iScience, 25(6):104393, 2022. ISSN 2589-0042.

---

> > ### Comment · Reviewer_5Kio · 2023-08-18
> > **Thank you for your rebuttal.**
> >
> > I have read the rebuttal and acknowledge the fact that the authors have conducted experimental results to validate their theoretical findings. I have updated my score and positively hope the paper is accepted.

---

### Official Review · Reviewer_sQGQ · 2023-08-02

**Soundness:** 4 excellent
**Presentation:** 4 excellent
**Contribution:** 4 excellent
**Rating:** 9
**Confidence:** 4

**Summary:**

The first contribution of this work is formalizing and analyzing a model of Unsupervised Machine Translation (UMT). The model applies even to low-resource source languages with massive domain gap and linguistic distance. Second, the paper exhibits two simple complementary models for which it is proved that:
(a) more complex languages require less common ground, and
(b) data requirements are not significantly greater than those of supervised translation (which tend to be significantly less than generative language modeling). These findings may have implications for the quantity and type of communication data that is collected for deciphering animal communication and for UMT more generally.

**Strengths:**

This work is the first theoretical work formally proving error bounds for prior-assisted translation, which shows that the sample complexity should remain roughly the same between the supervised and unsupervised settings (specifically, the two stylized models of language -- Random sentence trees and Random knowledge graphs), barring computational constraints.

**Weaknesses:**

There isn't any empirical evaluation aside from using text-davinci-02 to evaluate sentence likelihoods

**Questions:**

What has changed in this submission from the arxiv version (https://arxiv.org/pdf/2211.11081.pdf)?

---

> ### Author Rebuttal · Authors · 2023-08-10
>
> Thank you for your enthusiastic review and recognition of the importance of our findings, and  more generally of introducing a theoretical UMT framework.
>
> > There isn't any empirical evaluation aside from using text-davinci-02 to evaluate sentence likelihoods
>
> Thank you for this great suggestion. To address this, we have added synthetic experiments to validate our theoretical results. See the pdf attached to our overall response.

---

> > ### Comment · Reviewer_sQGQ · 2023-08-11
> > **The supplementary experiments are great**
> >
> > However, in Figure 3, the n_source = 4 case didn't run through as many samples as the other cases.

---

> > > ### Author Response · Authors · 2023-08-11
> > > **Number of samples in Figure 3**
> > >
> > > Thank you for carefully looking at our figure! That is correct and expected: when n_source=4 there are less samples than when n_source=7, which in turn has less samples than n_source=10.
> > >
> > > This is because, in the knowledge graph model, a sample from the source language $\mu$ is an edge in the source graph. When n_source = 4, there are $4^2=16$ possible directed edges. When n_source=7, there are $7^2 = 49$ possible directed edges, and when n_source=10 there are $10^2=100$ edges. In each experiment, we feed these edges (in random order) into the MLE algorithm, and show how the error decreases. Once all edges have been fed, there are no more possible samples, and so we stop the experiment. Since there are less possible edges for n_source=4 as compared to n_source=7,10, these experiments use less samples.
> > >
> > > (Pedantic note: Actually, $(\mathrm{n_{source}})^2$ is only an upper-bound on the number of edges, but is almost never the exact amount. This is because the marginal probability of any edge being present in a particular source graph is $p=0.5$.)

---

> > > > ### Comment · Reviewer_sQGQ · 2023-08-11
> > > > **thumbs up**
> > > >
> > > > Thank you for the explication and the analysis. I wish this work can get in and followed up by a broader audience, on the basis of the interesting theory developed for unsupervised machine translation.

---

### Author Rebuttal · Authors · 2023-08-10

# Overall Response

**BREAKING NEWS**: We have added experiments as requested by all reviewers. These synthetic experiments validate the theoretical findings and help illustrate the main message, that more complex languages can be translated with lower error. The focus of these experiments, like our paper, is on understanding how the error rate relates to the amount of data and language complexity, which may be useful for the scientists collecting data on animal communication. Additionally, we hope these experiments will help provide further intuition about our results.

Our experiments correspond to the two main theorems, 3.2 and 3.4. The results are in the attached pdf. We now describe in the experiments in detail, which are faithful brute-force implementations of the algorithms described in the paper.

**Common Nonsense model:** Since in this model the structure of sentences is arbitrary, we represent sentences by integer ids, $[10^5]=\{1,2,\ldots, 10^5\}$ and $[10^6]$ for the target language. We generate a prior $\rho$ from the common nonsense model by taking the target sentence ids $[10^6]$ and labeling a random $\alpha$-fraction of them as nonsense; the remaining sentences are called sensical $S$. Given a ground-truth translator $f_\star \colon [10^5] \to [10^6]$, the source language then distributes uniformly over the back-translation of sensical sentences, $f_\star^{-1}(S)$.

The translator family $\lbrace f_\theta | \theta \in \Theta\rbrace$ is taken to be a set of $10^5$ random one-to-one translators, of which one is secretely chosen to be ground-truth $f_\star$. We then train an MLE algorithm on random samples from the source language: Each sample $x\sim \mu$ rules-out a subset of translators, namely, all $\theta \in \Theta$ such that $f_\theta(x) \notin S$, i.e., is nonsensical.

Our experiments (Figure 1) show that as the number of samples increases, the average error over the plausible translators (that have not been ruled-out) decreases.

To show how language complexity / common ground affect translatability, we ablate the parameter $\alpha$ which determines the fraction of common nonsense. Our experiments validate the intuition that increased common nonsense results in lower translation error.

The error of a translator is computed as the fraction of disagreements with the ground-truth on a hold-out validation set of size 1000. The values with which the model is instantiated are:
| Name                        | Symbol   | Value(s)        |
|-----------------------------|----------|-----------------|
| Num source sentences        | $\|T\|$  | $10^5$          |
| Num target sentences        | $\|P\|$  | $10^6$          |
| Num training data           | $m$      | 1,...,100       |
| Num validation data         |          | 1000            |
| Fraction of common nonsense | $\alpha$ | 0,0.1,...,0.8   |

The experiments were run in parallel on an AWS r6i.4xlarge for a total of four CPU-hours.

**Knowledge Graph model:** Recall that in the knowledge graph model, text describes relations between nodes in a directed graph. Due to computational constraints, we consider ten nodes, each corresponding to a different word in the target language. To generate edges corresponding to the target language $P$, two nodes are connected with a directed edge independently, with probability $0.5$. We then consider source languages with $r \leq 10$ words. Given a ground-truth translator $f_\star \colon [r] \to [10]$, the source language graph $T$ is obtained by choosing a random subset of nodes $S$ of size $r$, taking the pre-image of graph induced on $S$ under $f_\star$, and (3) adding noise by redrawing each edge with probability $1-\alpha$ for a fixed _agreement coefficient_ $\alpha \in (0,1)$.

The prior $\rho$ is derived from the edges of $P$, and the source language $\mu$ is derived from the (noisy) permuted subgraph $T$. We consider the translator family $\lbrace f_\theta | \theta \in \Theta\rbrace$ of all node-to-node (word-to-word) injective translators, of which one is secretely chosen to be ground-truth. Similarly to the previous setting, we train an MLE algorithm on randomly chosen edges from $T$, which correspond to sentences in the source language. For each sampled edge $(x_1, x_2)$, we increase the "score" of each translator that agrees with the edge, that is, that $(f_\theta(x_1), f_\theta(x_2))$ is an edge in the graph $P$.

Our experiments (Figures 2 and 3) show that as the number of samples increases, the error of the top-scoring translator decreases.

To show how common ground affects translatability, we ablate the parameter $\alpha$ determines the fraction of edges on which the source language graph $T$ and the target language graph $P$ agree. Our experiments (Figure 2) validate the intuition that increased agreement results in lower translation error.

To show how language complexity affects translatability, we ablate $r$, which is the size of the subraph corresponding to the source language. Our experiments (Figure 3) validate the intuition that a larger subgraph results in lower translation error.

The error of a translator is computed as the fraction of edges whose labels are different than the ground-truth. The values with which the model is instantiated are:
| Name                        | Symbol   | Value(s)        |
|-----------------------------|----------|-----------------|
| Num source words        | r  | $1,4,7,10$          |
| Num target words        | $\|P\|$  | $10$          |
| Num training data           | $m$      | 1,2,..., up to all edges       |
| Edge density         |    p      | 0.5            |
| Agreement | $\alpha$ | $0, 0.33, 0.66, 1$ |

Note: We do not consider the product of all three ablations: Rather, for ablations on $r$ we take $\alpha=0.5$, and for ablations on $\alpha$ we take $r=9$.
The experiments were run in parallel on an AWS r6i.4xlarge for a total of three CPU-hours.

---

### Comment · Area_Chair_wnb1 · 2023-08-18
**note to authors**

I would like to thank the authors for their thoughtful responses. They will be taken into account.

---

### Decision · Program_Chairs · 2023-09-21

**Decision:**

Accept (poster)

**Comment:**

There has been a lot of debate about this paper in the reviewing process, and I believe the authors have defended relatively well the utility of their work as a formal model to show unsupervised translation is possible between pairs of languages under certain constraints and assumptions. While the excitement and reviews of the paper are somewhat mixed, it has some strong support.

I would be wary of making the connection to animal communication so explicit and visibly highlighted. As far as I can see, there is not much discussion in the paper of research about animal communication, and it serves as a highly informal (but perhaps catchy) inspiration.

I hope that the extra space, if the PCs eventually accept the paper, will be used for the synthetic experiments provided in the author's response, coupled with an intuition of how they support the formal model.

minor type in abstract: "sylized" should be "stylized?"